# Inferring Visual Biases in UAV Videos from Eye Movements

**Anne-Flore Perrin [1,2],\*** ⓘ**, Lu Zhang [2]** ⓘ **and Olivier Le Meur [1]** ⓘ

[1]  Univ Rennes, CNRS, IRISA, 263 Avenue Général Leclerc, 35000 Rennes, France; olemeur@irisa.fr
[2]  Univ Rennes, INSA Rennes, CNRS, IETR—UMR 6164, 35000 Rennes, France; lu.ge@insa-rennes.fr
**\***  Correspondence: anne-flore.perrin@irisa.fr; Tel.: +33-2-9984-2573

**Abstract:** Unmanned Aerial Vehicle (UAV) imagery is gaining a lot of momentum lately. Indeed, gathered information from a bird-point-of-view is particularly relevant for numerous applications, from agriculture to surveillance services. We herewith study visual saliency to verify whether there are tangible differences between this imagery and more conventional contents. We first describe typical and UAV contents based on their human saliency maps in a high-dimensional space, encompassing saliency map statistics, distribution characteristics, and other specifically designed features. Thanks to a large amount of eye tracking data collected on UAV, we stress the differences between typical and UAV videos, but more importantly within UAV sequences. We then designed a process to extract new visual attention biases in the UAV imagery, leading to the definition of a new dictionary of visual biases. We then conduct a benchmark on two different datasets, whose results confirm that the 20 defined biases are relevant as a low-complexity saliency prediction system.

**Keywords:** visual salience; Unmanned Aerial Vehicles (UAV); typical videos; visual attention; eye tracking; surveillance; biases; center bias

## 1. Introduction

Visual attention is the mechanism developed by the Human Visual System (HVS) to cope with the large quantity of information it is presented with. This capacity to sort out important information and to discard others can be represented thanks to visual salience representations. Saliency maps show the statistical distribution of human eye positions when seeing visual stimuli. There are many factors influencing where we look at, such as the visual content, the task at hand, and some systematic tendencies. We investigate the latter point in this paper in the specific context of Unmanned Autonomous Vehicles (UAVs) videos.

When watching natural scenes on a screen, observers tend to look at the center irrespective of the content [1,2]. This center bias is attributed to several effects such as the photographer bias [2], the viewing strategy, the central orbital position [3], the re-centering bias, the motor bias [4,5], the screen center bias [1], and the fact that the center is the optimized location to access most visual information at once [1]. It is usually represented as a centered isotropic Gaussian stretched to the video frame aspect ratio [6,7]. Other biases are present in gaze deployment, mainly due to differences in context, i.e., observational task [8], population sample [9], psychological state [10], or content types [11–13]. Biases are studied through various modalities, such as reaction times for a task [8,9], specifically designed architecture [9] or through hand-crafted features describing salience or fixations [11]. Note that, except for the center bias, we are not aware of visual attention biases that take the form of saliency patterns.

UAV imagery is getting momentum with the multiplication of potential applications it offers, i.e., new delivery services, journalism [14], and security and surveillance systems [15–17]. This imaging

differs from conventional contents on various aspects. For instance, the photographer bias is not the same due to the special training and visual routines required to control the aerial sensor [18,19]. Shot images and videos represent objects under a new and unfamiliar birds' perspective, with new camera and object motions, among others [20]. We thus wonder whether such variations impact visual explorations of UAV contents.

A first element of answer is proposed in previous works [21,22] that put into question the presence of center bias in UAV videos. We also claim that existing saliency models fail to predict saliency in UAV contents [21] because they are trained on unrepresentative ground truths. We thus aim to show a difference in attentional allocation between human saliency maps of typical and UAV sequences. To do so, we define a set of handcrafted features to represent UAV human saliency maps in a high-dimensional space and to discriminate them.

We then propose to dig further the question to understand and find out visual attention biases in UAV videos. Our aim is to produce a dictionary of biases possibly fed as priors to static or dynamic saliency prediction models. It is a widespread solution to improve saliency models, and it can take several forms. For instance, in [23], the dynamic two-streamed network includes a handmade dictionary of 2D centered-Gaussians which provides different versions of the center bias. To the best of our knowledge, it is the first time that one empirically extracts a dictionary of biases for UAV videos.

Our biases extraction follows a similar pipeline than a current attempt to analyze saliency patterns in movie shots to explore cinematography dynamic storytelling [24]. Their pipeline includes the definition of features (saliency maps predicted by Deep Gaze II [25]), a dimension reduction operation through a principal component analysis, and finally clustering using K-means.

Regarding the UAV ecosystem, deep learning is a mainstream tool for numerous applications, i.e., automatic navigation of UAV [19,26,27], object [27] or vehicle tracking [28–30], and (moving) object detection [31,32] under real-time constraints [33]. Some works combine both object detection and tracking [34–37] or implement the automation of aerial reconnaissance tasks [38]. However, only a minority of works take benefits from Regions of Interest (ROI) [27,39], sometimes in real time [40], which is a first step towards considering visual attention. We believe that saliency, and in particular a dictionary of biases, will enable the enhancement of current solutions, even in real-time applications.

In the remainder of the paper, we describe in Section 2 the material and followed methodology. It includes a description of conventional and UAV video datasets in Section 2.1, as well as the handcrafted features extracted from human saliency and fixation maps to represent visual salience in a high dimensional space in Section 2.2. We then verify the clear distinction between conventional and UAV videos saliency-wise using data analyses in Section 2.3. Interestingly, the used T-distributed Stochastic Neighbor Embedding (t-SNE) [41] representation presents differences within UAV saliency patterns. On this basis, biases in UAV videos are designed in Section 2.4. A benchmark is presented in Section 2.5 on two datasets—EyeTrackUAV1 and EyeTrackUAV2—to evaluate the relevance and efficiency of biases based on typical saliency metrics. Results are provided in Section 3 and discussed in Section 4. Finally, Section 5 concludes on the contributions of this work.

## 2. Materials and Methods

### 2.1. Datasets

Visual salience is an active field that tackles various issues such as segmentation [42], object recognition [43–45], object and person detection [46,47], salient object detection [48,49], tracking [50,51], compression [52,53], and retargeting [54]. Accordingly, numerous eye tracking datasets have been created for typical images [6,55–58]. To a lesser extent, it is possible to find datasets on videos [23,59–61]. Eventually, datasets on specific imaging (e.g., UAV videos) are getting growing attention and are being developed [20,22,62].

In this study, we consider only gaze information collected in free-viewing conditions. Observers can explore and freely appreciate the content.

### 2.1.1. Typical Videos

Among available datasets, the DHF1K [23] is the largest one for dynamic visual saliency prediction. It is a perfect fit for developing saliency models as it includes 1000 ($640 \times 360$) video sequences covering a wide range of scene variations. A 250 Hz eye tracking system recorded the gaze of 17 observers per stimuli under free-viewing conditions. We used the videos available in the training set (i.e., 700 videos), together with their fixation annotations. We have computed the human saliency maps following the process described in [22] to have comparable maps with the UAV videos dataset.

### 2.1.2. UAV Videos

In this regard, this study investigates the 43 videos of EyeTrackUAV2 (30 fps, $1280 \times 720$ and $720 \times 480$) [22], the largest and latest public gaze dataset for UAV content. Gaze information were recorded at 1000 Hz on 30 participants visualizing content extracted from datasets DTB70 [63], UAV123 [64], and VIRAT [65]. It presents a wide range of scenes, in terms of viewing angle, distance to the scene, among others. Besides, EyeTrackUAV2 was created in view to provide both free-viewing and surveillance-based task conditions. There are indeed contents compliant with objects detection and tracking, and contents with no salient object. We used fixation and human saliency maps generated from binocular data under free-viewing conditions, as recommended in the paper.

Last, in order to prove the external validity of results, we include the EyeTrackUAV1 dataset [20] in the benchmark of biases. This dataset includes 19 sequences coming from UAV123 ($1280 \times 720$ and 30 fps). Saliency information was recorded on 14 subjects under free-viewing conditions at 1000 Hz. Overall, the dataset comprises eye tracking information on 26,599 frames, which represents 887 s of video.

### 2.2. Definition and Extraction of Hand-Crafted Features

Extracting features from gaze information has a two-fold role. First, it enables the representation of saliency maps in a high dimensional space. Thanks to this representation, we expect to discriminate types of imaging and show the importance of developing content-wise solutions. Second, we also expect to find out specific characteristics, that would reveal discrepancies in gaze deployments occurring on conventional and UAV content. Those specific characteristics could be used to approximate content-wise biases. Note that in such case only features that can parameterize biases will be kept. We extract statistics for each frame based on the four representations, described below.

1.　Human Saliency Maps (HSMs)
2.　Visual fixation features
3.　Marginal Distributions (MDs) of visual saliency
4.　2D K-means and Gaussian Mixture Models (GMM) of fixations

### 2.2.1. Human Saliency Maps

Saliency maps are 2D topographic representations highlighting areas in the scene that attract one's attention. HSMs result from the convolution of the observers' fixations with a Gaussian kernel representing the foveal part of our retina [66]. In the following, a human saliency map at time $t$ is defined as $I_t : \Omega \subset \mathcal{R}^2 \to \mathcal{R}^+$, where $\Omega = [1 \ldots N] \times [1 \ldots M]$ with $N$ and $M$ the resolution of the input. Moreover, let $p_{\mathbf{i}}$ be the discrete probability of the pixel $\mathbf{i} = \langle x, y \rangle$ to be salient. The probability $p_{\mathbf{i}}$ is then defined as follows, such that $0 \leq p_{\mathbf{i}} \leq 1$ and $\sum_{\mathbf{i} \in \Omega} p_{\mathbf{i}} = 1$:

$$p_{\mathbf{i}} = \frac{I_t(\mathbf{i})}{\sum_{\mathbf{j} \in \Omega} I_t(\mathbf{j})} \tag{1}$$

From human saliency maps, we extract overall spatial complexity features and short-term temporal features as described below.

**Energy:** The energy of a pixel is the sum of the vertical and horizontal gradient absolute magnitudes. A Sobel filter of kernel size 5 is used as derivative operator. We define as features the average and standard deviation (std) of $E_t$ over all pixels of a frame.

$$E_t(\mathbf{i}) = |\frac{\partial I_t(\mathbf{i})}{\partial x}| + |\frac{\partial I_t(\mathbf{i})}{\partial y}| \tag{2}$$

High energy mean would indicate several salient regions or shape-wise complex areas of interest, whereas low energy would indicate more simple-shaped or single gaze locations.

**Entropy:** Shannon entropy is then defined as follows.

$$sE = -\sum\nolimits_{\mathbf{i}\in\Omega} p_{\mathbf{i}} log(p_{\mathbf{i}}) \tag{3}$$

A high value of entropy would mean that the saliency map contains a lot of information, i.e., it is likely that saliency is complex. A low entropy indicates a single zone of salience.

**Short-term temporal gradient:** We used a temporal gradient to characterise the difference over time in visual attention during a visualization. This gradient is computed as follows, with $G_{t_{max}}(\mathbf{i}) = 0$,

$$G_t(\mathbf{i}) = |I_t(\mathbf{i}) - I_{t+1}(\mathbf{i})| \tag{4}$$

A large gradient transcribes a large movement between $I_t$ and $I_{t+1}$.

### 2.2.2. Visual Fixation Features

We retrieve fixations from eye positions performing a two-step spatial and temporal threshold, the Dispersion-Threshold Identification (I-DT) [20,67]. Spatial and temporal thresholds were selected to be equal to 0.7 degree of visual angle and 80 ms, respectively, according to [68].

The fixation number can be representative of the number of salient objects in the scene. Their position may also indicate congruence between subjects [11,69]. We thus include **the number of fixations** per frame as features as well as **the number of clusters** derived from fixation positions.

The number of clusters is computed through two off-the-shelf typical clustering techniques [70,71], namely the Density-Based Spatial Clustering of Applications with Noise (DBSCAN) [72] and the Hierarchical extension of it (HDBSCAN) [73]. Regarding settings, we set the minimum number of points per cluster to 2. Other parameters were left to default values in python libraries (hdbscan and sklearn), such as the use of the euclidean distance.

### 2.2.3. Marginal Distributions

A marginal distribution describes a part of a multi-variable distribution. In our context, MDs represent vertical and horizontal salience, i.e., the sum of relative saliency of pixels over columns ($p_v$) and rows ($p_h$), respectively. MDs are computed as follows.

$$p_h(x) = \sum_{y\in\{1..M\}} p_{\mathbf{i}}(x,y), \qquad p_v(y) = \sum_{x\in\{1..N\}} p_{\mathbf{i}}(x,y) \tag{5}$$

Several illustrations of horizontal and vertical marginal distributions of UAV HSMs are provided in Figure 1 to qualitatively support the claim that the center bias does not necessarily apply to UAV contents. Accordingly, several descriptors of marginal distributions bring knowledge on horizontal and vertical saliency, which enables to study whether the center bias applies to UAV videos. Moreover, these parameters may be of capital importance when designing UAV parametric biases based on experimental data. From horizontal and vertical MDs, we extract the following features.

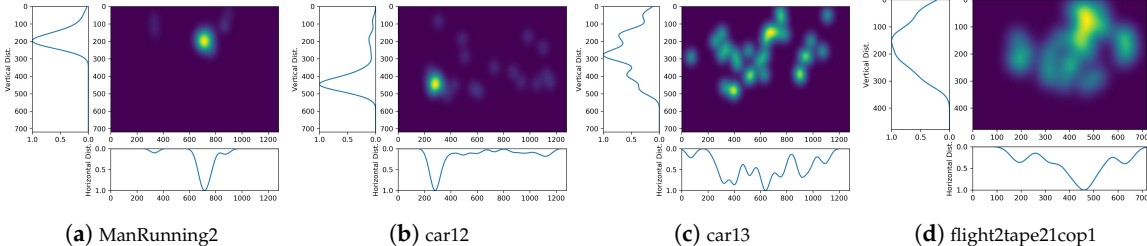

**(a)** ManRunning2      **(b)** car12      **(c)** car13      **(d)** flight2tape21cop1

**Figure 1.** Human Saliency Maps and their Marginal distributions of frame 165 for four sequences of EyeTrackUAV2.

**Moments:** empirical mean, variance, skewness, and kurtosis. Moments characterise the shape of a distribution. According to Fisher and Pearson's works [74], we express the $k^{th}$ moment as follows,

$$\mu^k = \frac{1}{n} \sum_{x=0}^{n} (p_{md}(x) - \bar{p})^k \tag{6}$$

where $p_{md} \in \{p_h, p_v\}$ is the marginal distribution, $n \in \{N, M\}$ is the number of samples in $p_{md}$, and $\bar{p}$ is the empirical mean of the distribution.

**Median and geometric mean:** A Gaussian distribution mean is equal or similar to its geometric mean and median. Thus, these two values help characterizing further marginal distributions, as well as indicate "how far" from a 1D single-Gaussian distribution marginal distributions are.

### 2.2.4. 2D K-Means and Gaussian Mixture Models

Single and multiple clusters are formed from fixation positions. We use K-means and GMMs to approximate the features of these groups. On the one hand, a single cluster brings knowledge about general behavior. On the other hand, having several batches allows for a more precise data depiction. Besides, the group expressing the largest variance is particularly interesting to express saliency when opposed to a single cluster. We thus ran both algorithms to approximate a single cluster as well as multiple classes. We characterize only the most representative batch in the latter scenario.

**K-means** [75] is the most used clustering algorithm in machine learning due to its accuracy, simplicity, and low complexity. We define as feature the center position of a cluster. Two features are thus extracted, one from a single batch, the second from the most representative group—in terms of fixations number—of a multi-clustering. Regarding implementation details, HDBSCAN algorithm gives the number of classes, and all runs follow a random initialization.

**Gaussian Mixture Models:** Similarly to K-means, we consider a single Gaussian and the Gaussian in the GMM that accounts for the highest variance. Extracted features are their means, covariance matrices, and variance (i.e., the extent of fixation points covered by the Gaussian).

### 2.2.5. Overall Characteristics

We extract 38 characteristics per frame. Table 1 summarizes the features describing human saliency maps. When considering the entire sequence, means and standard deviations of these components are extracted temporally, i.e., over the entire set of frames. This gives 76 features per video, independently of the sequences' number of frames. As a side note, all features relative to a position have been normalized to be free from datasets resolutions.

**Table 1.** Number of features extracted per frame and per sequence.

| Description Level | Source Data | Features | nb Features |
|---|---|---|---|
| Frame | Saliency maps | Energy mean and std, entropy, temporal gradient mean and std | 5 |
| | Fixations | Number of fixations, DBSCAN and HDBSCAN number of clusters | 3 |
| | Marginal distribution | First to fourth degree moments, geometric mean and median of marginal distributions | 12 |
| | Gaussian mixture models | Center coordinates of K-mean algorithm ( with one and HDBSCAN number of clusters) together with means and covariances of the most representative 2D Gaussian of 1- and HDBSCAN-GMM, and proportion of variance explained by these Gaussians | 18 |
| Sequence | All | Mean and std of all precedent characteristics over time | 76 |

## 2.3. Visualization and Classifications

In this section, we want to verify whether attention in typical and UAV content exhibits different patterns. We already know that some sequences of EyeTrackUAV2, mainly originating from the VIRAT dataset, are likely to present a center bias and thus have similar HSMs than those of conventional videos. On the contrary, we expect other sequences, originating from DTB70 and UAV123, to present HSMs with different biases, as claimed in [21,22].

Before exploring and generating biases, we visualize data in a low-dimensional space, thanks to the t-SNE technique [41]. This process maps elements in a 2D space accordingly to the distribution of distances in the high-dimension space. It is a current technique to classify and discriminate elements [76,77]. The t-SNE is advantageous for two reasons: it enables visualization, and most importantly, it comprehensively considers all dimensions in its representation. In our context, the t-SNE shows distance differences between saliency maps based on extracted features and discriminates content-based peculiarities.

If the t-SNE algorithm clusters typical HSMs separately from UAV HSMs, per sequence and frame, this attests that there are different visual attention behaviors in UAV imaging. Otherwise, there is no point in performing biases identification. This is a crucial step towards content-wise biases creation as we first verify there are distinctions between conventional and UAV HSMs and thus justify the existence of such biases. Second, we can determine clusters in which saliency characteristics are similar and learn biases based on the t-SNE 2D distance measure.

### 2.3.1. Discrimination between Salience in Typical and UAV Content

As recommended in [78], we paid close attention to hyperparameters selection of the t-SNE method. Therefore we verified several perplexity values (e.g., 2, 5, 10, 15, 20, 30, and 50) for each t-SNE run. To stay in line with previous settings, the minimum of components is 2 and distances are euclidean. Any t-SNE starts with random initialization, runs with a learning rate of 200, and stops after 1000 iterations.

The process was applied on all UAV contents of EyeTrackUAV2 and the 400 first videos of the DHF1K dataset. This represents an overall number of 275,031 frames, 232,790 for typical and 42,241 for UAV content. We considered all features when dealing with frames, while we computed the temporal average and std of feature values when dealing with sequences. In addition, we ran the t-SNE on Marginal Distribution (MD) parameters only, to verify whether they have the abilities of a separability criterion. If so, they can help to reconstruct biases.

### Results

All results follow similar tendencies over all tested parameters. They are described and illustrated in Figure 2 with perplexity 30. In Figure 2a, we added the thumbnails of the first frame of sequences for more readability. Colors were attributed to the points, or to the thumbnails, depending on the dataset that elements are originating from. In Figure 2a, we can observe the spread of HSMs in the space, per sequence and considering all features. We note that UAV videos (colored) are located moderately apart from conventional videos, especially these coming from UAV123 (red) and DTB70 (green). Furthermore, most of UAV VIRAT videos (blue) are closer to typical sequences than other UAV content. These sequences are possibly sharing saliency characteristics with certain conventional videos. This fact

supports previous claims, in that the majority of VIRAT HSMs present a center bias. It is also striking that traditional videos are separated into two large clusters. This could be further explored, though not in this paper as it would get out of the scope of our study.

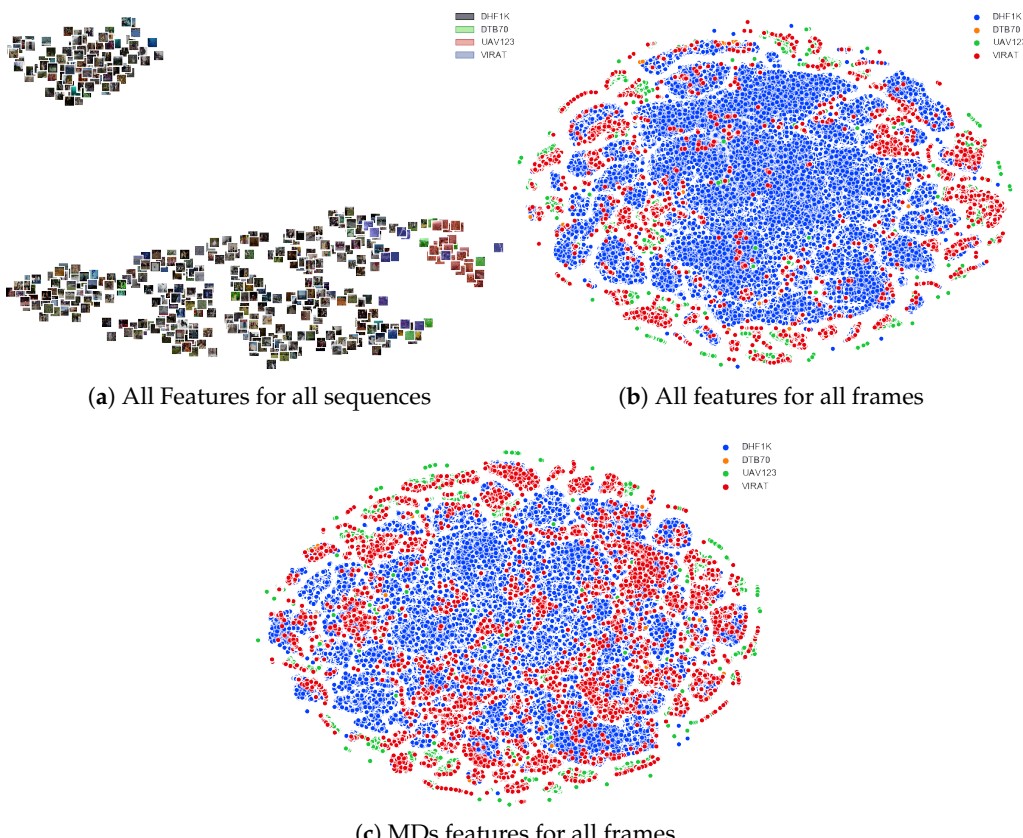

(**a**) All Features for all sequences　　　　(**b**) All features for all frames

(**c**) MDs features for all frames

**Figure 2.** t-SNE learnt on all features and marginal distribution characteristics, on sequence and frame data (perplexity values 30). We can see that most points representing UAV content are in areas of the space not covered by typical points, and that there are various clusters of UAV points.

Figure 2b depicts how HSMs spread spatially for each frame, based on all extracted features. We can observe a huge set of close clusters toward the center of the space, containing mainly DHF1K frame representations, with a fair proportion of VIRAT points and some UAV123 maps. At the borders, we can observe a lot of small batches, formed mainly by UAV contents. These clusters are more distant to their neighbors than those in the center of the space. That is, it makes sense to study various biases in UAV videos as there are numerous batches of UAV content representations clearly distinct from other groups.

Figure 2c presents the dispersion of HSMs, per frame and based on marginal features only. As for the previous observations (see Figure 2b), we observe a distinction between representations of typical and UAV content, with UAV clusters at the borders and more distant to other batches. A difference stands in that we find more VIRAT points within centered groups.

Based on the three illustrations, and considering the generalization of these results over other values of perplexity and features sets, we can proceed with the identification of biases in UAV content. It is also reasonable to rely only on marginal distributions characteristics to generate these biases.

2.3.2. Clustering of UAV Sequences

In this section, the focus is set on UAV sequences only as we aim to derive the attentional biases. In other words, from t-SNE representations, we aim to identify clusters of different gaze deployment habits in UAV visualizations. Initially, biases are regarded as generic behaviors that are found consistently in visual attention responses. Here, we specifically want to extract prevailing and UAV-specific biases.

In pilot tests, we have tried generating a dictionary of biases via a systematic variation of parameters of an anisotropic 2D Gaussian distribution. We also created a dictionary with the human means of the 43 sequences of EyeTrackUAV2. Results, comprising usual similarity metrics of the saliency field applied on biases and HSMs, showed weak performances. These studies emphasized the need to derive patterns of salience from HSMs directly. Additionally, parameters on which we cluster HSMs must be used for the construction of parametric biases. That points out the exclusion of some previous handcrafted features, such as statistics on HSMs and fixation features, hardly useful to generate parametric biases. Besides, we want biases free from any constraints, which excludes 2D K-means and GMM predictions that require prior knowledge. Thus, we focus on marginal distributions features to classify UAV sequences. Moreover, the previous section has emphasized their relevance.

We thus performed a t-SNE with a low perplexity to derive our clusters based only on MDs parameters on EyeTrackUAV2 sequences. Perplexity can be seen as a loose measure of the effective number of neighbors [41]. We thus selected a perplexity of 5 to have about eight clusters. It is a compromise between being content-specific and deriving meaningful and prominent patterns. We also ran several iterations of t-SNE, with perplexity 2, 10, and 20, to ensure the robustness of the obtained classification.

In line with previously defined settings, we relied on the 2D-Euclidean distance in the t-SNE representation to compute the similarity between samples. We computed a confusion matrix based on a hierarchical clustering algorithm (dendrogram) implemented with the Ward criterion [79,80]. The Ward variance minimization algorithm relies on the similarity between clusters centroïds. Hierarchical clustering is highly beneficial in our context as it is less constrained than K-means. There is no need for prior knowledge, such as a number of classes, number of elements in groups, or even whether batches are balanced.

Results

Following the tree architecture, sequences showing a Ward dissimilarity under 150 are forming clusters. We ultimately determine seven classes to learn content-wise biases. Results are compiled in Figure 3 and Table 2. Figure 3a presents this distance matrix between sequences HSMs. Figure 3b shows the t-SNE spatial dispersion of sequences and outlines the seven groups. Human Means (HMs) thumbnails allow verifying the similarity between sequences HSMs in clusters.

Overall, most classes show homogeneous human means. For instance, in group II, there is a recurrent vertical and thin salient pattern. The most heterogeneous batch would be cluster IV, for which the averaging process used to generate HMs may not be an optimal temporal representation. Finally, Table 2 describes clusters, their number of sequences, and the number of frames it embodies. To make up for the hardly readable axes on Figure 3a, Table 2 follows the order of sequences used to form the confusion matrix.

**Table 2.** Clusters, their sequences, number of sequence, and frame.

| I | II | III | IV | V | VI | VII |
|---|---|---|---|---|---|---|
| 6 seq. 4891 frames | 7 seq., 10694 frames | 2 seq., 613 frames | 7 seq., 8932 frames | 8 seq., 9549 frames | 5 seq., 2435 frames | 8 seq., 5127 frames |
| Soccer1 | ManRunning2 | Girl1 | ManRunning1 | 09152008flight2tape1_3_crop1 | building1 | Girl2 |
| 09152008flight2tape1_3_crop2 | car4 | Walking | car14 | 09152008flight2tape2_1_crop2 | car11 | Basketball |
| truck3 | wakeboard8 | | 09162008flight1tape1_1_crop2 | 09152008flight2tape1_3_crop3 | person22 | bike2 |
| 09152008flight2tape3_3_crop1 | car3 | | 09152008flight2tape1_5_crop1 | 09152008flight2tape1_5_crop2 | building3 | building2 |
| car13 | car9 | | 09152008flight2tape2_1_crop4 | StreetBasketball1 | truck2 | truck4 |
| car15 | car1 | | car7 | building4 | | car12 |
| | car2 | | 09162008flight1tape1_1_crop1 | bike3 | | Soccer2 |
| | | | | 09152008flight2tape2_1_crop1 | | 09152008flight2tape2_1_crop3 |

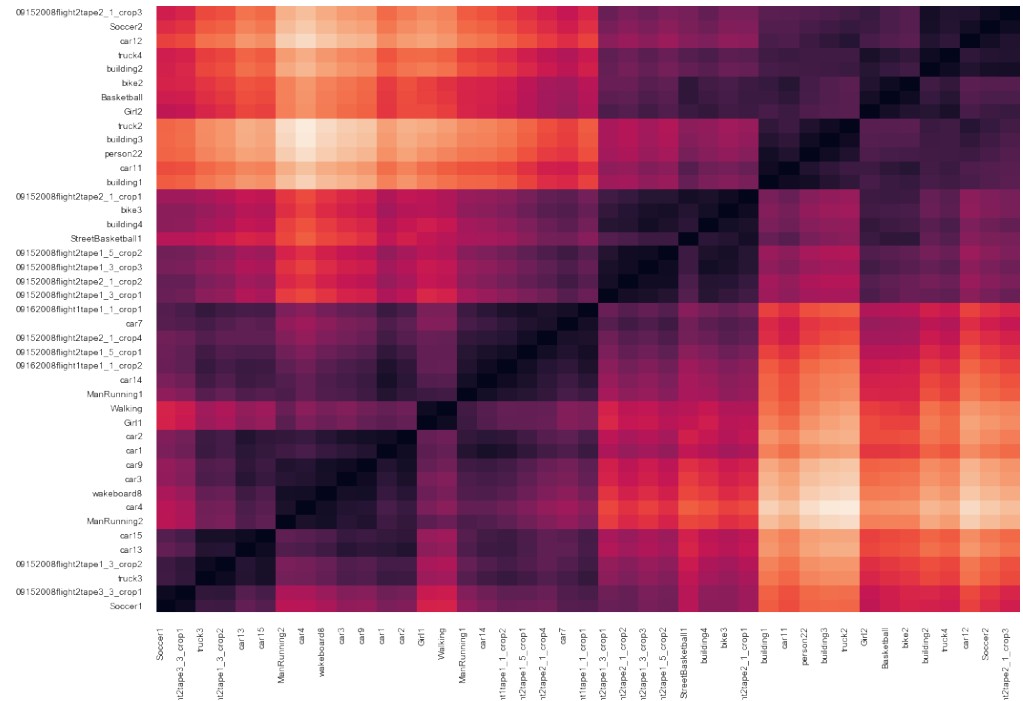

(**a**) t-SNE 2D Euclidean Distance-confusion Matrix

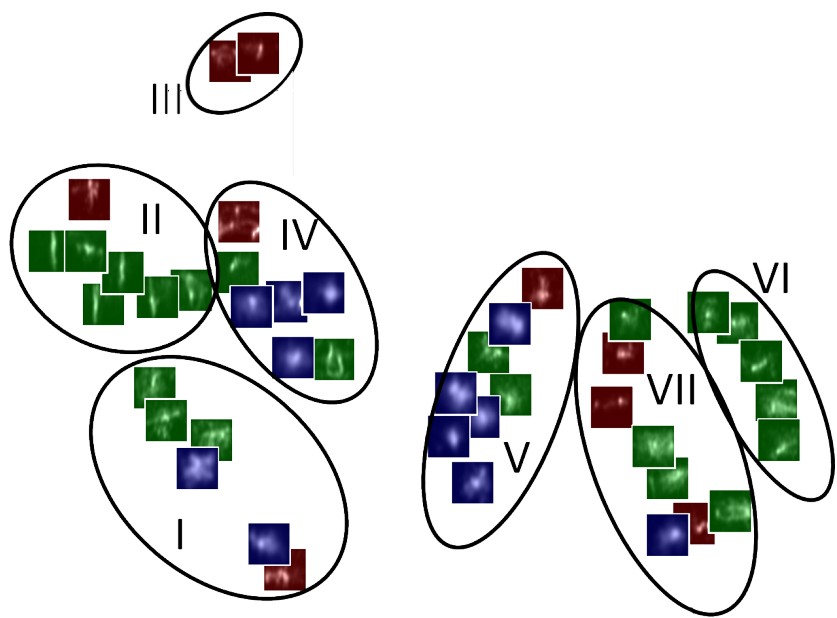

(**b**) t-SNE, p=5, threshold=150

**Figure 3.** Cluster selection from t-SNE.

## 2.4. Extrapolation of Biases

This section introduces the process designed to generate parametric saliency biases for UAV imaging. First, we need to define which properties of HSM ground truth will be exploited to parameterize and generate biases. From the set of such features, we need to derive generic and prominent patterns. The entire process is described below and is summarized in Figure 4. At this point, sequences have been clustered based on their saliency marginal distribution features.

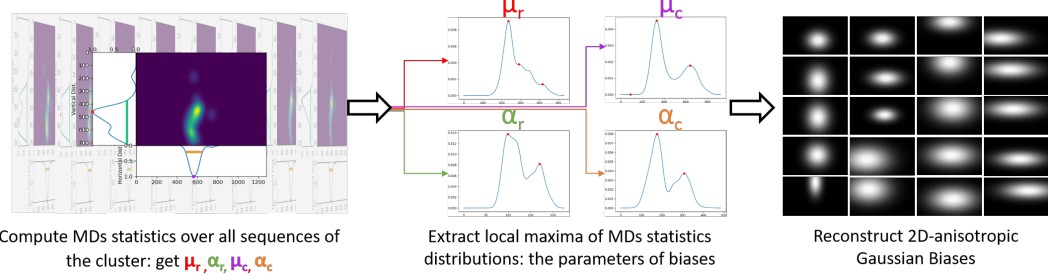

Compute MDs statistics over all sequences of　　Extract local maxima of MDs statistics　　Reconstruct 2D-anisotropic
the cluster: get $\mu_r$, $\alpha_r$, $\mu_c$, $\alpha_c$　　distributions: the parameters of biases　　Gaussian Biases

**Figure 4.** Pipeline describing the biases generation process. First, we compute vertical and horizontal marginal distribution of a frame, from which we extract means ($\mu$) and standard deviations ($\sigma$). Variations of these four features are studied through their distributions, over all frames of sequences in a cluster. The peaks of these distributions are extracted as biases parameters. Eventually, by combining values of peaks, we create a dictionary of biases. This process is applied on each cluster individually.

### 2.4.1. Extraction of MDs Statistics

The idea of extracting biases roots in the prevalence of a center bias in generic audiovisual stimuli salience. The center bias is often represented by a centered isotropic Gaussian stretched to the video frame aspect ratio [6,7]. Here, we get free from the isotropic and centered characteristics of the Gaussian. Moreover, the stretching may not apply to UAV contents, which, for instance, often present smaller size objects. Under these considerations, biases are to be parameterized by a set of two mean and standard deviation parameters, representing the Gaussian center and spreading, respectively. Such 1D-Gaussian parameters are derived from MDs of the ground truth human saliency maps. That is, we compute mean and std values of the marginal distributions, horizontally and vertically, for each HSM frame of each sequence, i.e., $\mu_r$, $\sigma_r$, $\mu_c$, and $\sigma_c$, respectively. We will refer to these values as MDs statistics, to make the difference with MDs features.

Doing so, we assume that MDs follow 1D-Gaussian distributions horizontally and vertically. As we can see in Figure 1, this claim is reasonable if there is a behavior similar to single-object tracking (i.e., Figure 1a,b). However, it is more questionable when there is no object of interest (i.e., Figure 1c). Still, we believe this choice is sound for the following reasons.

- Averaging, and to a lesser extent getting the standard deviation, over the HSM columns and rows act as filtering unrepresentative behaviors.
- It is seldom to observe several areas of interest within a single HSM. Often, the congruence between observers is pretty low when it happens. For instance, such events may occur when there is a shift of attention from one object to another. Biases do not target to capture such behaviors. Besides, if there is no object of interest, we expect some biases to present center-bias alike patterns to deal with it.
- Plans are to use biases as a dictionary, and particularly to combine these derived salience patterns. This fact should cope with some of the inaccuracies made during this process.
- Last but not least, we are dealing with video saliency, i.e., a frame visualization lasts about 0.3 s. Assuming a 1D-Gaussian marginal distribution for a HSM, we make a rather small error. However, pooling these errors could reveal to be significant. Accordingly, the significance of error depends more on the strategy for biases parameters extraction from MDs statistics. We discuss this issue in the following section.

### 2.4.2. Local Maxima of MDs Statistics Distributions

The identification of biases parameters from MDs statistics is critical as it could introduce or exacerbate errors in priors predictions. To study patterns in MDs, the stress is laid on clustering, which serves the examination of prominent patterns within saliency-wise similar content. Consequently,

the investigation of biases patterns relies on distributions of MDs statistics over each cluster. Figure 5 presents the obtained distributions, named MDS-D.

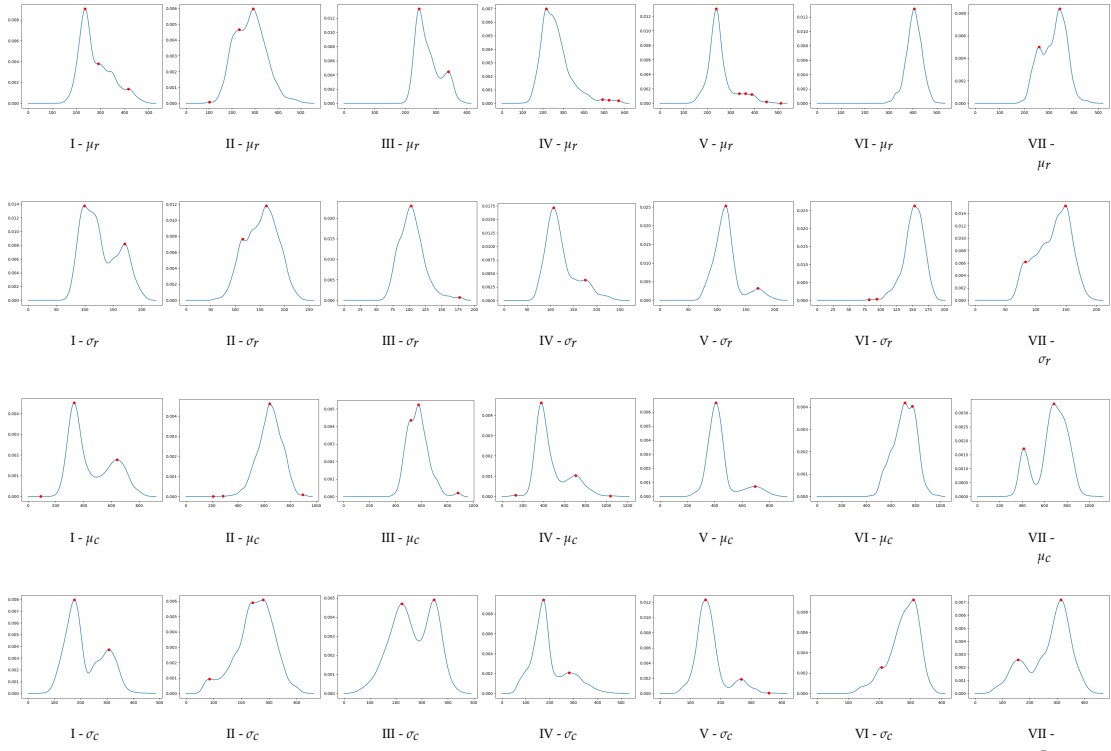

**Figure 5.** Examples of MDs statistic distributions and the extracted peak values. Statistics are means ($\mu$) and standard deviation ($\sigma$) per columns (c) or rows (r). Associated values can be found in Table 3.

**Table 3.** Position (in pixel) of local maxima in distributions of statistic values. These peaks are the future parameters necessary for biases generation. Highest peaks are highlighted in bold.

| Cluster | I | II | III | IV | V | VI | VII |
|---|---|---|---|---|---|---|---|
| $\mu_r$ | **236**, 291, 416 | 102, 231, **292** | **245**, 340 | **217**, 491, 523, 570 | **238**, 335, 361, 388, 449, 510 | **405** | 259, **343** |
| $\sigma_r$ | **99**, 170 | 115, **163** | **103**, 177 | **107**, 175 | **115**, 171 | 82, 94, **153** | 83, **149** |
| $\mu_c$ | 91, **332**, 640 | 210, 284, **643**, 896 | 519, **577**, 880 | 131, **376**, 705, 1038 | **409**, 695 | **712**, 772 | 415, **683** |
| $\sigma_c$ | **176**, 307 | 85, 241, **279** | 223, **346** | **174**, 282 | **150**, 267, 358 | 207, **309** | 158, **315** |
| Number of biases | 36 | 72 | 24 | 64 | 72 | 12 | 16 |

It is straightforward that usual statistics over MDS-Ds, such as the mean, will not make meaningful, prevailing, and accurate bias parameters. Notions of preponderance, precision, and significance directly involve the computation of local maxima in distributions. Local maxima, also referred to as peaks, embed the aspect of likelihood, can be computed precisely, and give a value that is directly useful for biases generation.

To conduct a robust computation of local maxima, we estimate the probability density function of MDS-Ds using a Gaussian kernel density estimation (kde). Extrapolating the distribution exhibits two main advantages. First, we conduct a biases parameters extraction based on likelihood. This particularly fits the aim of biases. Second, probability density functions are less prone to noise, which could have dramatic results in extrema extractions, especially direct comparison of neighboring values.

We found important to include all local maxima, even if it means including less relevant future bias parameters. Besides, there is no strong basis to rely on regarding setting a threshold of meaningfulness of a peak. For instance, $II - \sigma_c$ shows a peak at 241, and similarly $III - \mu_c$ presents a maximum at 880, which would have been rejected with a precisely set threshold despite their likely

importance. Accordingly, a post-biases-creation filtering is needed to select the most relevant biases (see Section 3.1).

In order to create a comprehensive and exhaustive bank of biases, all possible combinations of MDS-D parameters are considered in the process. It means that in cluster I, there will be $\#\mu_r \times \#\sigma_r \times \#\mu_c \times \#\sigma_c = 3 \times 2 \times 3 \times 2 = 36$ patterns. Table 3 reports the values of peaks extracted from the distributions. It also indicates how many biases were generated per cluster.

### 2.4.3. Reconstruct Parametric 2D-Anisotropic Gaussian Biases

To construct a 2D-anisotropic Gaussian pattern, we need four parameters: the coordinates of the center of the Gaussian, and horizontal and vertical standard deviations. A bias is thus defined by the 2D-Gaussian of horizontal mean $\mu_c$ and std $\sigma_c$, and vertical mean $\mu_r$ and std $\sigma_r$. It forms a $1280 \times 720$ image resulting from the dot product of the 1D-Gaussian distributions formed horizontally and vertically (see Equations (7) and (8)). Bicubic interpolation is used to resize biases when dealing with VIRAT sequences, which have a smaller resolution.

$$Bias(\mathbf{x}, \mathbf{y}, \mu_r, \sigma_r, \mu_c, \sigma_c) = \mathbf{G}(\mathbf{x}, \mu_r, \sigma_r)^T * \mathbf{G}(\mathbf{y}, \mu_c, \sigma_c) \tag{7}$$

$$\text{with } \mathbf{G}(\mathbf{x}, \mu, \sigma) = \frac{1}{\sigma\sqrt{2\pi}} \exp(-\frac{(\mathbf{x} - \mu)^2}{2\sigma^2}) \tag{8}$$

where $\mathbf{x} = \langle 1 \dots 720 \rangle$, $\mathbf{y} = \langle 1 \dots 1280 \rangle$, and $T$ is the transpose operator. The formed patterns are calculated for each cluster, for each comparison of biases parameters. The dictionary reaches an overall number of 296, from which a subset is presented in Figure 6.

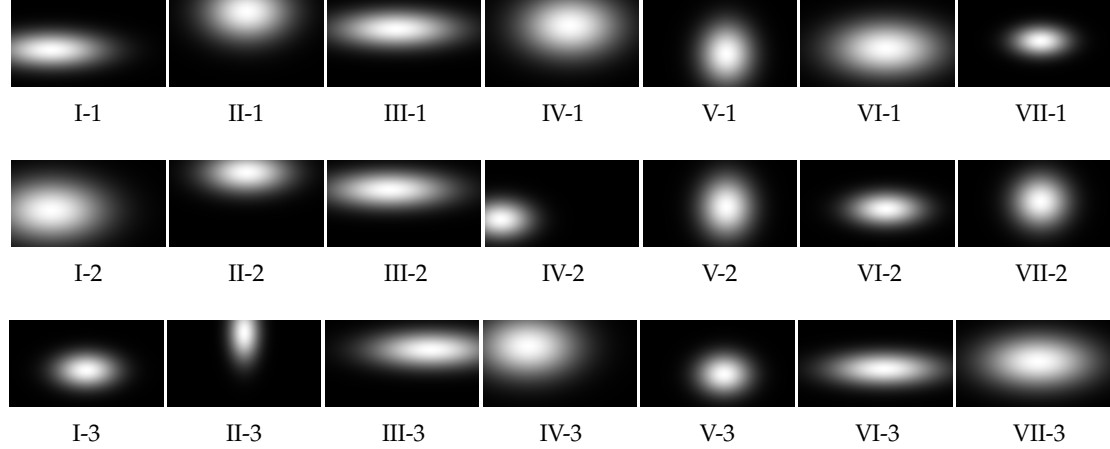

**Figure 6.** Examples of a bias extracted per cluster

### 2.5. Biases Analyses

We conduct several analyses to verify that extracted biases are meaningful and representative of human deployment of visual attention in UAV videos. First, we examine the relative similarity between biases and the EyeTrackUAV2 dataset, overall, and per cluster. Results sort out biases by efficiency and lead to the selection of a reduced and reliable dictionary. It follows a qualitative analysis of the obtained dictionary.

Then, we aim to situate the efficiency of biases when compared to baselines and hand-crafted features saliency models. Additionally, we verify the external validity of the performance of extracted biases on EyeTrackUAV1. We thus conduct the benchmark on both EyeTrackUAV1 and EyeTrackUAV2.

Finally, we explore an additional function of the dictionary: improving static handcrafted features saliency models by filtering their results with biases. The results of thes analyses will precise the potential usages of the dictionary.

2.5.1. Benchmark Metrics

All along this analysis, we employ typical saliency metrics. We made the decision to exclude metrics involving fixations on the basis that fixation extraction processes may interfere with the results. We want to get as free as possible from a possible dataset bias. Accordingly, we included three state-of-the-art saliency metrics recommended in the MIT benchmark [66,81]: Pearson's Coefficient of Correlation (CC), Similarity (SIM), and the Kullback Leibler divergence (KL).

**CC**: The range of the correlation metric goes from $-1$ to 1, 1 representing a perfect fit between the data.

**SIM**: Similarity measures a perfect fit between the data histograms as 1, and no similarity as 0.

**KL**: This metric exhibits a dissimilarity score, emphasising the error made during the prediction. It thus favors patterns with large variance in order to reduce the amount of error made. The score ranges from 0 to infinity ($+\infty$), with having the lowest score the better.

Each metric is computed on every frame. Frame scores are then averaged over sequences, cluster or entire dataset. Although not being optimal regarding temporal considerations, it is the most widely used practice to evaluate dynamic salience [23,82–86].

2.5.2. Benchmark Multi-Comparisons

The aim of biases is to perform better than a conventional Center Bias (CB), and would ideally challenge handcrafted features saliency models (HCs) prediction accuracy. This section introduces the CB, HMs, and HCs stimuli compared to biases in the benchmark.

**The center bias** is a centered isotropic Gaussian stretched to video frame aspect ratio [7]. CB is a popular prior used in typical imaging. It thus sets a perfect baseline to measure biases.

**Human means** are the temporally averaged saliency maps over an entire sequence. It gives 19 HMs for EyetrackUAV1, and 43 for EyetrackUAV2.

**Handcrafted features saliency models**: Based on the work in [21], the selected HCs depict the range of prediction accuracy of most typical hand-crafted features saliency models. BVS [87], GBVS [88], and RARE2012 [89] being the most predictive models, SIM [90] and SUN [91] the least.

2.5.3. Handcrafted Features Saliency Models Filtered with Biases

Filtering HCs with biases will provide new insights for the power of prediction of biases and the information it brings. To sustain a low-complexity constraint, we define the filtering operation as the Hadamard product between the bias and the prediction map of an HC (see Equation (9)).

$$FilteredHC = HC \circledast Bias \tag{9}$$

Moreover, we measure the efficiency of this process through gain which is the difference between scores of filtered HC and original HCs or biases results.

$$Gain_X = score(FilteredHC) - score(X) \tag{10}$$

with $X \in \{HC, Bias\}$, and score is the outcome of CC, SIM, or KL measures.

## 3. Results

### 3.1. A First Quantitative Analysis: Selection of Biases

In this section, we sort out biases per cluster to select the most performing ones and build a reliable dictionary. We have selected three of most predictive patterns per cluster, under the constraint that they outperform the CB, and that they are dissimilar enough. Efficiency was considered for the three metrics simultaneously, favoring biases achieving the best results for the three of them. Sorting reduces

down to 3 × 7 the number of saliency patterns in the dictionary. From now on, we name biases the 21 patterns selected to form the dictionary.

Table 4 reports the parameters of biases and their respective metric results on EyeTrackUAV2 overall and per cluster. We emphasized the results standing beyond the others, per cluster.

**Table 4.** Parameters of biases and saliency metric results of biases on EyeTrackUAV2. Bold numbers achieve the highest score per cluster.

| | Biases Parameters | | | | Overall | | | Per Cluster | | |
|---|---|---|---|---|---|---|---|---|---|---|
| | $\mu_r$ | $\sigma_r$ | $\mu_c$ | $\sigma_c$ | CC ↑ | SIM ↑ | KL ↓ | CC ↑ | SIM ↑ | KL ↓ |
| CenterBias | | | | | **0.305** | **0.349** | **1.553** | 0.104 | 0.238 | 2.423 |
| I_1 | 416 | 99 | 332 | 307 | 0.122 | 0.258 | 3.335 | **0.309** | **0.325** | 2.215 |
| I_2 | 416 | 170 | 332 | 307 | 0.100 | 0.261 | 2.110 | 0.278 | 0.292 | **1.727** |
| I_3 | 416 | 99 | 640 | 176 | 0.282 | 0.333 | 3.526 | 0.291 | 0.307 | 3.627 |
| CenterBias | | | | | **0.305** | **0.349** | **1.553** | 0.201 | 0.163 | 2.687 |
| II_1 | 102 | 163 | 643 | 241 | 0.083 | 0.253 | 2.790 | 0.299 | 0.229 | **2.097** |
| II_2 | 102 | 115 | 643 | 241 | 0.022 | 0.203 | 6.507 | 0.285 | 0.235 | 2.374 |
| II_3 | 102 | 163 | 643 | 85 | 0.095 | 0.207 | 10.033 | **0.306** | **0.258** | 4.601 |
| CenterBias | | | | | **0.305** | **0.349** | **1.553** | 0.147 | 0.284 | 1.846 |
| III_1 | 245 | 103 | 577 | 346 | 0.196 | 0.304 | 2.912 | 0.270 | 0.355 | **1.540** |
| III_2 | 245 | 103 | 519 | 346 | 0.177 | 0.296 | 2.990 | 0.265 | **0.357** | 1.587 |
| III_3 | 245 | 103 | 880 | 346 | 0.194 | 0.303 | 3.044 | **0.272** | 0.323 | 1.622 |
| CenterBias | | | | | **0.305** | **0.349** | **1.553** | -0.004 | 0.132 | 3.676 |
| IV_1 | 217 | 175 | 705 | 282 | 0.231 | 0.320 | 1.672 | -0.020 | **0.147** | **3.518** |
| IV_2 | 491 | 107 | 131 | 174 | -0.075 | 0.095 | 14.913 | **0.096** | 0.132 | 14.501 |
| IV_3 | 217 | 175 | 376 | 282 | 0.079 | 0.256 | 2.282 | -0.007 | 0.146 | 4.836 |
| CenterBias | | | | | **0.305** | **0.349** | **1.553** | 0.306 | 0.300 | 1.635 |
| V_1 | 449 | 171 | 695 | 150 | 0.260 | 0.325 | 3.310 | 0.469 | 0.387 | 1.613 |
| V_2 | 388 | 171 | 695 | 150 | 0.297 | 0.344 | 3.083 | 0.453 | 0.383 | **1.608** |
| V_3 | 449 | 115 | 695 | 150 | 0.253 | 0.315 | 4.354 | **0.484** | **0.404** | 2.236 |
| CenterBias | | | | | 0.305 | **0.349** | 1.553 | 0.379 | 0.417 | 1.181 |
| VI_1 | 405 | 153 | 712 | 309 | **0.314** | 0.340 | **1.527** | **0.474** | 0.442 | **1.009** |
| VI_2 | 405 | 94 | 712 | 207 | 0.299 | 0.344 | 3.024 | 0.470 | **0.477** | 1.912 |
| VI_3 | 405 | 94 | 712 | 309 | 0.288 | 0.337 | 2.553 | 0.464 | 0.466 | 1.388 |
| CenterBias | | | | | 0.305 | 0.349 | 1.553 | 0.430 | 0.352 | 1.327 |
| VII_1 | 343 | 83 | 683 | 158 | 0.301 | 0.341 | 4.606 | **0.592** | **0.488** | 2.522 |
| VII_2 | 343 | 149 | 683 | 158 | 0.314 | **0.355** | 2.718 | 0.521 | 0.412 | 2.024 |
| VII_3 | 343 | 149 | 683 | 315 | **0.335** | 0.351 | **1.460** | 0.502 | 0.355 | **1.287** |

**All biases perform the highest on their cluster** without exception. It confirms the interest to use these biases as content-based priors and is highly promising for the future regarding the use of the dictionary of biases in saliency studies.

Regarding the expectations we had for biases, **patterns of cluster IV show low prediction scores**. They are always under our previsions, though higher than the CB. We went through the content to understand this behavior. It turns out that videos of class IV present a lot of movements, and meteorologic conditions interfere with two sequences. A single image, such as bias, is not able to capture patterns related to extreme camera and object movements or other impacting conditions.

On the contrary, **biases from clusters V, VI, and VII exceed our expectations**. These classes' biases go beyond the purpose set for them. In addition to outperforming the CB per cluster, they are more reliable overall.

**Clusters I, II, and III reach the expected efficiency**, with a specific mention for III which outreaches our expectations for SIM and KL metrics.

Regarding overall results, some biases do not show a better predictive power than the CB. On the other hand, as already mentioned, all saliency patterns outperform CB

cluster-wise. This confirms that **the clustering and biases generation processes designed to derive content-specific behavior are sound**. This is also supported by the fact that content from classes II, III, and IV are hardly predicted by other biases than their respective ones.

To further discuss our clustering strategy, the results hint to pool classes V, VI, and VII. This would be reasonable because the three groups have the same parent in the hierarchical dendrogram set in Section 2.3.2. However, combining all sequences would have an impact on the diversity of the patterns in the dictionary.

### 3.2. A Qualitative Analysis

Figure 6 illustrates the selected saliency patterns. Most of the extracted biases of the last three clusters, especially for VI and VII, can be seen as **variations of a center bias more in line with UAV content peculiarities**. For instance, objects are smaller and could accordingly necessitate precise saliency areas. This is reflected in salience patterns with centered 2D Gaussian with low std in at least one direction.

More than half of biases present a 2D Gaussian with a high longitudinal std, which looks like human attitudes towards 360° contents [92]. Observers may act towards high altitude UAV contents as they would do for omnidirectional content.

In clusters II and III, and to a lesser extent in class IV, saliency is specifically seen at the top of the content. We make a connection between this fact and the UAV sequence itself. **The position of objects of interest**, in this context in the upper part of the content, **strongly influences visual attention**. An upward camera movement may strengthen this fact. However, this latter fact alone seems less significant because other videos with such camera motion show distinct patterns (such as truck3).

Note that bias II-3 is highly representative of videos peculiarities. Indeed, we can observe a vertical Gaussian accounting for the vertical road depicted in most videos of this cluster.

Finally, in classes II, V, VI, and VII, and to a lesser extent in III, **biases are centered width-wise**. Moreover, at least one bias per cluster presents this characteristic. To the best of our knowledge, it is the first overall behavior exhibited from analyses on gaze deployment in UAV imaging.

### 3.3. Biases Benchmark

In this section, we aim to situate the efficiency of biases when compared to baselines and handcrafted features saliency models. We also provide an evaluation of the true power of prediction of biases, both on their own and coupled with saliency prediction models. Additionally, we verify the external validity of the performance of extracted biases on EyeTrackUAV1. We thus conduct the benchmark on both EyeTrackUAV1 and EyeTrackUAV2.

#### 3.3.1. Results on EyeTrackUAV2

#### CB, HMs, and HCs

Results obtained by biases are presented in Table 4, and those of baselines and HCS in Table 5. Biases not only compete with the CB but also surpass it, per cluster. Some of them exhibit a center-bias alike pattern suiting more to UAV peculiarities. The biases extracted from clusters V, VI, and VII present less dispersed 2D Gaussians. These patterns seem to cope with the high distance between the UAV and the scene, i.e., the relatively small size of objects of interest and a longitudinal exploration of content.

**Table 5.** HC models results overall and per cluster, as well as HM average results, for EyeTrackUAV2. Stressed results show the best results per metric per cluster.

| | Overall | | | I | | | II | | | III | | | IV | | | V | | | VI | | | VII | | |
|---|---|---|---|---|---|---|---|---|---|---|---|---|---|---|---|---|---|---|---|---|---|---|---|---|
| | CC↑ | SIM↑ | KL↓ | CC↑ | SIM↑ | KL↓ | CC↑ | SIM↑ | KL↓ | CC↑ | SIM↑ | KL | CC↑ | SIM↑ | KL↓ | CC↑ | SIM↑ | KL↓ | CC↑ | SIM | KL↓ | CC↑ | SIM↑ | KL↓ |
| BMS | 0.266 | 0.313 | 1.550 | 0.350 | 0.285 | 1.664 | **0.448** | 0.226 | 2.091 | **0.462** | 0.329 | 1.448 | **0.381** | 0.267 | 1.937 | 0.194 | 0.254 | 1.816 | **0.436** | 0.407 | 1.050 | **0.376** | 0.292 | **1.647** |
| GBVS | **0.327** | **0.337** | **1.447** | 0.236 | 0.265 | 1.827 | 0.360 | 0.187 | 2.222 | 0.439 | **0.350** | **1.341** | 0.274 | 0.228 | 2.140 | **0.274** | 0.272 | **1.707** | 0.350 | 0.387 | 1.156 | 0.284 | 0.274 | 1.663 |
| RARE2012 | 0.298 | 0.324 | 1.598 | **0.366** | **0.311** | **1.589** | 0.376 | 0.213 | 2.158 | 0.377 | 0.345 | 1.466 | 0.347 | 0.267 | 2.026 | 0.236 | 0.266 | 1.881 | **0.438** | **0.415** | **1.043** | 0.240 | 0.285 | 1.774 |
| SIM | 0.124 | 0.270 | 1.760 | 0.261 | 0.249 | 1.798 | 0.198 | 0.125 | 2.703 | 0.312 | 0.270 | 1.697 | 0.191 | 0.179 | 2.352 | 0.119 | 0.219 | 2.014 | 0.210 | 0.344 | 1.295 | 0.037 | 0.216 | 2.050 |
| SUN | 0.155 | 0.280 | 1.741 | 0.197 | 0.236 | 1.907 | 0.113 | 0.114 | 2.906 | 0.213 | 0.243 | 1.812 | 0.152 | 0.172 | 2.427 | 0.143 | 0.221 | 2.009 | 0.280 | 0.362 | 1.219 | 0.033 | 0.217 | 2.123 |
| HM | 0.220 | 0.306 | 3.054 | 0.117 | 0.224 | 4.535 | 0.087 | 0.129 | 4.407 | 0.085 | 0.232 | 3.444 | 0.005 | 0.119 | 6.670 | 0.239 | **0.274** | 2.826 | 0.300 | 0.375 | 2.715 | 0.329 | **0.320** | 2.058 |

HMs are particularly unsuitable for clusters II and IV. Moreover, the struggle to predict the saliency of these two classes is a revealing recurrent issue, stressing the importance of biases. HMs predictions are low for I and III and mild for the last three classes. For obvious reasons, HMs of VIRAT, showing center-bias alike maps, achieve the highest scores. When compared to HMs, biases III-1, IV-1, V, VI, and VII show superior overall prediction efficiency. Cluster-wise, biases achieve a better score, including cluster IV. Biases, with no exception, are more relevant for salience prediction than HMs, which is a fine achievement. Besides, no behavior such as having a HM performing particularly well on its cluster has been observed. One reason to this observation could be that HMs are too specific to the content to have a power of generalization over a cluster, even more over an entire dataset.

Now that biases outrun baseline maps, we deal with static saliency models that use handcrafted features to make predictions. Such comparisons measure whether biases capture relevant behaviors that reach the efficiency of more elaborate and refined models.

Over the entire dataset, biases from clusters VI and VII, and to a lesser extent V and III, outreach the scores of HCs. On the contrary, biases extracted from classes I, II, and IV hardly compete. That makes sense since the former clusters exhibit similar patterns to center bias, which has a strong power of generalization over a dataset. Though, the latter ones extracted more content-related behaviors, which poorly express general salience. Concerning clusters, V, VI, and VII show better scores for biases. To a lesser extent, biases of I, II, and III outdo SIM and SUN. Only biases from IV have a prediction power far worse than HCs.

The takeaway message is that several priors have a high overall prediction power able to compete with more complex HCs. Biases are efficient content-wise and surpass at least SIM and SUN. It implies they go beyond envisioned baselines and carry out relevant saliency information. It is thus highly interesting to use biases as low complexity saliency predictions.

Handcrafted Models Filtered with Biases

Table 6 gives the obtained results. Overall, filtering exacerbates the results obtained before. Predictions on clusters V, VI, and VII are improved and present the highest gains. Biases I-3, III-all, and IV-1 are slightly less impacting filters. II-all and IV-3 only make some mild improvements. Unsurprisingly, filtering with II-all and IV-3 do not perform well overall. Their performance over specific sequences though justifies their use. For instance, II-all presents the highest gains in SIM and CC for all HC on sequences DTB70 ManRunning2, UAV123 car9, and car11, among others.

This brings us to note that biases are particularly efficient on the sequences which constitute their cluster. Moreover, for at least one of these videos, the obtained gain is the highest when compared to other biases. Besides, an impressive outcome is the presence of this improvement on the other sequences.

**Improving saliency results demonstrate that biases bring new knowledge about saliency to HCs which is beneficial to bring up the efficiency of predictions.** The increase of accuracy is nevertheless rather low. Table 6 presents the overall gain per model, which is representative of the range obtained sequence-wise. Obviously, there are more benefits to use biases on less accurate models, such as SIM and SUN. Doing so achieves an improvement of about 0.1 in CC. It is less interesting to use filtering on better models, with an overall gain approaching 0.01 in CC for RARE2012 and BMS. This difference is quite more significant than expected. Finally, the filtered SUN does not reach the accuracy of unfiltered BMS. Thus, the advantage of using biases as filters is not encouraging.

This does not call into question the meaning and interest in the dictionary. But, **it discards its use as a bank of filters.**

**Table 6.** Saliency metric gain$_{HC}$ for HC models filtered with biases and CB per cluster on EyeTrackUAV2. Result showing a gain (positive for CC and SIM, negative for KL) are highlighted for more readability.

| | CC↑ | SIM↑ | KL↓ | CC↑ | SIM↑ | KL↓ | CC↑ | SIM↑ | KL↓ | CC↑ | SIM↑ | KL↓ | CC↑ | SIM↑ | KL↓ | CC↑ | SIM↑ | KL↓ | CC↑ | SIM↑ | KL↓ |
|---|---|---|---|---|---|---|---|---|---|---|---|---|---|---|---|---|---|---|---|---|---|
| | **I-1** | | | **II-1** | | | **III-1** | | | **IV-1** | | | **V-1** | | | **VI-1** | | | **VII-1** | | |
| BMS | −0.052 | −0.021 | 1.567 | −0.080 | −0.018 | 0.989 | **0.008** | **0.022** | 1.200 | **0.059** | **0.048** | **−0.062** | **0.076** | **0.047** | 1.569 | **0.141** | **0.068** | **−0.207** | **0.076** | **0.044** | 2.929 |
| GBVS | −0.113 | −0.038 | 1.695 | −0.130 | −0.035 | 1.136 | **0.262** | −0.004 | 1.386 | −0.010 | **0.023** | 0.131 | **0.011** | **0.026** | 1.716 | **0.052** | **0.045** | **−0.024** | 0.000 | **0.013** | 3.133 |
| RARE2012 | −0.080 | −0.029 | 1.585 | −0.107 | −0.030 | 1.052 | **0.253** | **0.002** | 1.282 | −0.003 | **0.029** | 0.042 | **0.021** | **0.029** | 1.636 | **0.060** | **0.051** | **−0.115** | **0.014** | **0.020** | 3.002 |
| SIM | **0.020** | −0.008 | 1.518 | **0.006** | **0.001** | 0.922 | **0.225** | **0.042** | 1.105 | **0.145** | **0.065** | **−0.155** | **0.164** | **0.065** | 1.469 | **0.217** | **0.082** | **−0.295** | **0.191** | **0.075** | 2.778 |
| SUN | **0.004** | −0.011 | 1.531 | −0.020 | −0.009 | 0.956 | **0.227** | **0.034** | 1.138 | **0.114** | **0.056** | **−0.116** | **0.132** | **0.055** | 1.515 | **0.185** | **0.074** | **−0.252** | **0.157** | **0.062** | 2.833 |
| | **I-2** | | | **II-2** | | | **III-2** | | | **IV-2** | | | **V-2** | | | **VI-2** | | | **VII-2** | | |
| BMS | −0.041 | −0.010 | 0.323 | −0.165 | −0.076 | 4.705 | −0.007 | **0.014** | 1.273 | −0.297 | −0.200 | 13.180 | **0.100** | **0.062** | 1.367 | **0.096** | **0.057** | 1.313 | **0.109** | **0.070** | 1.018 |
| GBVS | −0.099 | −0.023 | 0.433 | −0.217 | −0.093 | 4.779 | **0.248** | −0.011 | 1.454 | −0.348 | −0.212 | 13.186 | **0.024** | **0.035** | 1.546 | **0.015** | **0.027** | 1.528 | **0.026** | **0.039** | 1.230 |
| RARE2012 | −0.066 | −0.015 | 0.345 | −0.187 | −0.088 | 4.722 | **0.244** | −0.003 | 1.347 | −0.310 | −0.205 | 13.109 | **0.033** | **0.039** | 1.454 | **0.034** | **0.034** | 1.402 | **0.037** | **0.044** | 1.121 |
| SIM | **0.011** | 0.000 | 0.293 | −0.063 | −0.052 | 4.629 | **0.034** | **0.034** | 1.183 | −0.180 | −0.172 | 13.106 | **0.197** | **0.083** | 1.252 | **0.196** | **0.081** | 1.185 | **0.210** | **0.093** | 0.891 |
| SUN | −0.003 | −0.004 | 0.305 | −0.090 | −0.062 | 4.653 | **0.210** | **0.026** | 1.215 | −0.208 | −0.178 | 13.099 | **0.162** | **0.072** | 1.299 | **0.162** | **0.070** | 1.242 | **0.175** | **0.081** | 0.942 |
| | **I-3** | | | **II-3** | | | **III-3** | | | **IV-3** | | | **V-3** | | | **VI-3** | | | **VII-3** | | |
| BMS | **0.077** | **0.045** | 1.801 | −0.114 | −0.079 | 8.300 | **0.001** | **0.019** | 1.339 | −0.072 | −0.018 | 0.512 | **0.053** | **0.030** | 2.620 | **0.100** | **0.054** | 0.837 | **0.144** | **0.076** | **−0.252** |
| GBVS | −0.002 | **0.016** | 1.998 | −0.168 | −0.099 | 8.400 | **0.264** | −0.003 | 1.499 | −0.133 | −0.036 | 0.652 | −0.015 | **0.006** | 2.767 | **0.015** | **0.027** | 1.033 | **0.051** | **0.048** | **−0.042** |
| RARE2012 | **0.019** | **0.023** | 1.875 | −0.138 | −0.091 | 8.292 | **0.246** | −0.002 | 1.431 | −0.094 | −0.026 | 0.548 | **0.002** | **0.011** | 2.674 | **0.034** | **0.034** | 0.916 | **0.057** | **0.055** | **−0.146** |
| SIM | **0.178** | **0.070** | 1.676 | −0.006 | −0.056 | 8.201 | **0.214** | **0.039** | 1.251 | −0.005 | −0.003 | 0.458 | **0.152** | **0.053** | 2.507 | **0.186** | **0.073** | 0.733 | **0.233** | **0.092** | **−0.352** |
| SUN | **0.145** | **0.058** | 1.733 | −0.038 | −0.066 | 8.222 | **0.223** | **0.032** | 1.274 | −0.026 | −0.009 | 0.481 | **0.120** | **0.042** | 2.555 | **0.156** | **0.064** | 0.778 | **0.199** | **0.083** | **−0.310** |
| | **CB** | | | | | | | | | | | | | | | | | | | | |
| BMS | **0.097** | **0.035** | **−0.162** | | | | | | | | | | | | | | | | | | |
| GBVS | **0.034** | **0.023** | **−0.069** | | | | | | | | | | | | | | | | | | |
| RARE2012 | **0.024** | **0.028** | **−0.116** | | | | | | | | | | | | | | | | | | |
| SIM | **0.156** | **0.042** | **−0.208** | | | | | | | | | | | | | | | | | | |
| SUN | **0.131** | **0.038** | **−0.187** | | | | | | | | | | | | | | | | | | |

To summarize, filtering with biases restrict the salience spatially. Such a constraint is often beneficial. Overall, filtering exacerbates biases effects, given several good general biases (i.e., patterns from V, VI, and VII) and several specific patterns (i.e., I-all, II-all, and VI-3). Accuracy results are directly linked to the complexity of the used models. Indeed, complex models show fewer improvements than a less elaborate predictor. Under this study conditions, using the dictionary as a bank of filters does not seem optimal. Still, results prove that biases bring new and advantageous saliency information on their own. This supports using the dictionary as a set of priors.

### 3.3.2. Results on EyeTrackUAV1

In order to verify the external validity of the bank of biases, we carry out the same study than above on another dataset, namely, EyeTrackUAV1.

### Biases, CB, HMs and HCs

Biases on EyeTrackUAV1 match our expectations overall and sequences-wise, especially when compared to results of [21]. Results—presented in Table 7 and for CC on sequences in Table 8—confirm that biases from clusters III, V, VI, and VII are good generic predictors, while those from I, II, and IV are more content-centric. For instance, II-all are efficient for person18. II-3 is particularly efficient for car13, perhaps because it draws a typical road pattern. Biases from I are accurate predictors on car2, I-3 being particularly applicable for building5. Last, IV reaches high scores for boat6, and car8, among others. Hence, some sequences are better represented by specific biases. These outcomes confirm the previous analyses on biases and recall the importance of clustering. The last point, KL continues favoring biases with high dispersion, meaning IV-1, VI-1, and VII-3. Note that based on its very poor effect on both datasets, **IV-2 is evicted from the dictionary**.

**Table 7.** Overall metric results of biases, CB, HMs, and HCs on the dataset EyeTrackUAV1. Highlighted results for biases are outperforming CB, stressed HC scores outdo the best bias.

| | I-1 | I-2 | I-3 | II-1 | II-2 | II-3 | III-1 | III-2 | III-3 | IV-1 | IV-2 | IV-3 | V-1 | V-2 | V-3 | VI-1 | VI-2 | VI-3 | VII-1 | VII-2 | VII-3 | CB | HMs | BMS | GBVS | RARE2012 | SIM | SUN |
|---|---|---|---|---|---|---|---|---|---|---|---|---|---|---|---|---|---|---|---|---|---|---|---|---|---|---|---|---|
| CC↑ | 0.103 | 0.104 | 0.221 | 0.127 | 0.070 | 0.154 | 0.215 | 0.204 | 0.157 | 0.221 | 0.058 | 0.130 | 0.201 | 0.249 | 0.175 | 0.220 | 0.208 | 0.190 | **0.277** | **0.283** | 0.268 | 0.229 | 0.223 | **0.383** | **0.373** | **0.335** | 0.216 | 0.197 |
| SIM↑ | 0.175 | 0.174 | **0.235** | 0.190 | 0.160 | 0.202 | 0.218 | 0.215 | 0.201 | 0.215 | 0.061 | 0.186 | 0.220 | 0.240 | 0.212 | 0.209 | 0.227 | 0.210 | **0.268** | 0.257 | 0.225 | 0.184 | 0.228 | 0.248 | 0.245 | 0.256 | 0.187 | 0.187 |
| KL↓ | 3.739 | 2.430 | 3.771 | 2.854 | 5.523 | 7.647 | 2.985 | 3.025 | 3.229 | **2.030** | 13.562 | 2.398 | 3.317 | 3.057 | 4.396 | **2.051** | 3.587 | 3.236 | 4.394 | 2.724 | **1.914** | 2.186 | 3.856 | **1.811** | **1.803** | **1.868** | 2.151 | 2.208 |

**Table 8.** CC results of biases, CB and HCs per sequence on the dataset EyeTrackUAV1. Highlighted results for biases are outperforming CB. Highest HC scores are also stressed.

| CC↑ | Bike3 | Boat6 | Boat8 | Building5 | Car10 | Car13 | Car2 | Car4 | Car6 | Car7 | Car8 | Group2 | Person13 | Person14 | Person18 | Person20 | Person3 | Truck1 | Wakeboard10 |
|---|---|---|---|---|---|---|---|---|---|---|---|---|---|---|---|---|---|---|---|
| I-1 | 0.067 | 0.038 | 0.164 | 0.172 | 0.102 | 0.116 | 0.180 | −0.039 | 0.149 | 0.066 | 0.147 | 0.086 | 0.062 | 0.140 | −0.046 | 0.151 | 0.104 | 0.232 | 0.058 |
| I-2 | 0.042 | 0.094 | 0.127 | 0.118 | 0.106 | 0.143 | 0.175 | −0.055 | 0.152 | 0.069 | 0.155 | 0.084 | 0.069 | 0.155 | −0.013 | 0.141 | 0.109 | 0.219 | 0.090 |
| I-3 | 0.292 | 0.078 | **0.435** | **0.344** | **0.331** | 0.214 | **0.227** | 0.145 | **0.316** | **0.197** | 0.169 | **0.277** | 0.096 | 0.153 | 0.039 | 0.155 | **0.306** | 0.229 | 0.197 |
| II-1 | 0.128 | **0.280** | 0.059 | −0.126 | 0.155 | 0.225 | 0.061 | **0.204** | 0.144 | 0.057 | 0.167 | 0.135 | 0.094 | 0.080 | **0.243** | 0.032 | 0.171 | 0.089 | 0.204 |
| II-2 | 0.051 | **0.234** | −0.040 | −0.157 | 0.065 | 0.181 | 0.028 | 0.168 | 0.088 | 0.011 | 0.124 | 0.062 | 0.032 | **0.220** | −0.006 | 0.083 | 0.002 | 0.117 | |
| II-3 | 0.148 | **0.298** | 0.151 | −0.038 | 0.240 | **0.300** | 0.016 | 0.149 | 0.198 | 0.082 | 0.149 | 0.163 | 0.088 | 0.070 | **0.204** | 0.023 | **0.256** | 0.106 | **0.323** |
| III-1 | 0.242 | **0.379** | 0.222 | −0.078 | **0.319** | 0.214 | 0.100 | 0.181 | 0.202 | 0.133 | **0.222** | **0.244** | 0.144 | **0.190** | **0.228** | 0.142 | **0.318** | **0.311** | **0.360** |
| III-2 | 0.214 | **0.378** | 0.204 | −0.082 | **0.301** | 0.205 | 0.098 | 0.149 | 0.196 | 0.125 | **0.222** | 0.226 | 0.140 | **0.189** | **0.220** | 0.145 | **0.303** | **0.298** | **0.344** |
| III-3 | 0.248 | **0.222** | 0.189 | −0.069 | 0.249 | 0.128 | 0.056 | **0.221** | 0.120 | 0.090 | 0.129 | 0.197 | 0.093 | 0.118 | **0.159** | 0.083 | 0.228 | 0.255 | **0.256** |
| IV-1 | 0.302 | **0.315** | 0.246 | −0.015 | **0.316** | **0.259** | 0.127 | **0.286** | 0.232 | 0.143 | **0.262** | 0.134 | | 0.158 | **0.242** | 0.108 | **0.304** | 0.249 | **0.314** |
| IV-2 | −0.120 | −0.073 | −0.080 | −0.026 | −0.101 | −0.044 | −0.045 | −0.103 | −0.073 | −0.067 | −0.033 | −0.067 | −0.028 | 0.005 | −0.102 | −0.006 | −0.082 | 0.018 | −0.074 |
| IV-3 | 0.075 | **0.282** | 0.095 | −0.080 | 0.166 | 0.168 | 0.106 | 0.008 | 0.151 | 0.066 | 0.202 | 0.121 | 0.106 | 0.145 | **0.172** | 0.138 | 0.177 | 0.186 | 0.192 |
| V-1 | 0.300 | 0.081 | **0.362** | **0.286** | **0.313** | 0.216 | 0.136 | **0.257** | **0.278** | **0.202** | 0.103 | **0.277** | 0.072 | 0.109 | 0.090 | 0.101 | **0.271** | 0.159 | 0.204 |
| V-2 | **0.351** | 0.160 | **0.427** | **0.249** | **0.391** | **0.266** | 0.142 | **0.311** | **0.315** | **0.219** | 0.151 | **0.342** | 0.109 | 0.148 | 0.138 | 0.123 | **0.358** | 0.226 | **0.300** |
| V-3 | 0.275 | 0.022 | **0.355** | **0.329** | 0.276 | 0.171 | 0.142 | **0.199** | **0.262** | 0.185 | 0.089 | **0.245** | 0.050 | 0.094 | 0.041 | 0.105 | 0.228 | 0.131 | 0.128 |
| VI-1 | **0.388** | 0.103 | **0.372** | **0.320** | **0.323** | 0.196 | 0.186 | 0.191 | **0.260** | **0.218** | 0.148 | **0.270** | 0.087 | 0.154 | 0.082 | 0.157 | **0.259** | 0.258 | 0.200 |
| VI-2 | **0.374** | 0.050 | **0.432** | **0.320** | **0.327** | 0.170 | 0.165 | 0.181 | **0.263** | **0.205** | 0.132 | **0.284** | 0.074 | 0.134 | 0.046 | 0.142 | **0.277** | 0.215 | 0.167 |
| VI-3 | **0.361** | 0.036 | **0.387** | **0.333** | 0.283 | 0.143 | 0.173 | 0.121 | 0.230 | 0.184 | 0.134 | 0.238 | 0.068 | 0.134 | 0.023 | 0.160 | 0.228 | 0.233 | 0.130 |
| VII-1 | **0.382** | 0.165 | **0.572** | 0.176 | **0.459** | 0.241 | 0.135 | **0.235** | **0.296** | **0.220** | 0.186 | **0.389** | 0.130 | **0.199** | 0.110 | 0.160 | **0.464** | **0.341** | **0.397** |
| VII-2 | **0.377** | **0.232** | **0.471** | **0.211** | **0.445** | **0.298** | 0.158 | **0.310** | **0.338** | **0.229** | 0.196 | **0.378** | 0.140 | 0.184 | **0.168** | 0.145 | **0.426** | **0.291** | **0.382** |
| VII-3 | **0.414** | **0.230** | **0.411** | **0.212** | **0.395** | 0.247 | 0.196 | **0.227** | **0.293** | **0.221** | **0.220** | **0.323** | 0.133 | **0.207** | 0.152 | **0.187** | **0.350** | **0.351** | **0.312** |
| CB | 0.340 | 0.210 | 0.280 | 0.209 | 0.299 | 0.253 | 0.203 | 0.194 | 0.259 | 0.194 | 0.204 | 0.241 | **0.214** | 0.184 | 0.158 | 0.176 | 0.240 | 0.282 | 0.218 |
| BMS | 0.181 | **0.653** | 0.263 | **0.395** | 0.375 | | **0.287** | **0.356** | 0.337 | 0.259 | 0.319 | 0.553 | 0.302 | 0.466 | **0.426** | 0.410 | **0.761** | 0.438 | 0.388 |
| GBVS | 0.241 | 0.542 | 0.334 | 0.330 | **0.406** | **0.160** | 0.159 | 0.251 | **0.359** | **0.267** | **0.344** | **0.566** | **0.364** | **0.474** | 0.409 | **0.412** | 0.679 | 0.365 | **0.420** |
| RARE2012 | **0.246** | 0.618 | **0.340** | 0.380 | 0.247 | 0.106 | 0.096 | 0.166 | 0.322 | 0.204 | 0.300 | 0.475 | 0.221 | 0.400 | 0.376 | 0.349 | 0.695 | **0.491** | 0.338 |
| SIM | 0.100 | 0.417 | 0.300 | 0.339 | 0.174 | 0.035 | 0.090 | 0.029 | 0.140 | 0.128 | 0.198 | 0.299 | 0.147 | 0.236 | 0.276 | 0.301 | 0.350 | 0.366 | 0.170 |
| SUN | 0.127 | 0.349 | 0.235 | 0.348 | 0.095 | 0.077 | 0.111 | 0.025 | 0.131 | 0.095 | 0.197 | 0.249 | 0.123 | 0.162 | 0.297 | 0.239 | 0.367 | 0.355 | 0.152 |

CB is less predictive than at least one bias for each content. Moreover, biases V-2, and VII outdo CB overall. This fact confirms the validity and interest of this dictionary.

HMs of respective sequences surpass biases, CB, and on most occasions HCs. However, such performance is restricted to their sequence. On other content, they barely compete with best biases sequence-wise. There are only two exceptions: HMs of car10 and person3. They present patterns highly located towards the center of the content (similar to II-3 and VII-2).

HCs results highly depend on the content. For instance, they outreach HMs and biases on boat6, car8, person14, person18, person20, person3, and truck1. Yet, they are less reliable than biases or HM on bike3, boat8, car10, car13, and wakeboard10. We could not relate the behavior of HCs and biases on these contents based on the annotations of the database.

Regarding EyeTrackUAV1 contents, Person13 is very hard to predict, for all tested maps. In a different vein, wakeboard10 is noticeable: its HM reaches a CC score of 0.84 of prediction, while biases and HCs hardy getting CC scores of 0.3. Accordingly, additional patterns could be added in the future to deal with sequences similar to these two in the future.

To sum up, biases show improvements for specific contents, not well predicted by HCs. Overall, biases reach the set expectations, validating the external validity of the dictionary.

Handcrafted Models Filtered with Biases

Scores obtained while filtering HCs with biases on EyeTrackUAV1 are reported in Table 9. Filters with the greatest dispersion—IV-1, VI-1, and VII-3—are the only ones to show a gain in KL, for all HC. This fact is counterbalanced by SIM results. They report progress for all biases for SIM and SUN, and only biases from II do not improve BMS, GBVS, and RARE2012. Thanks to CC scores, we are able to sort out biases from most to least performing: VII, VI, IV-1, and V, followed by I-2 and -3, III, and IV-3, and at the end I-1 and II biases.

**Table 9.** Saliency metric gain$_{HC}$ for HC models filtered with biases and CB overall on EyeTrackUAV1. Result showing a gain (positive for CC and SIM, negative for KL) are highlighted for more readability.

| | CC↑ | SIM↑ | KL↓ | CC↑ | SIM↑ | KL↓ | CC↑ | SIM↑ | KL↓ | CC↑ | SIM↑ | KL↓ | CC↑ | SIM↑ | KL↓ | CC↑ | SIM↑ | KL↓ | CC↑ | SIM↑ | KL↓ |
|---|---|---|---|---|---|---|---|---|---|---|---|---|---|---|---|---|---|---|---|---|---|
| | **I-1** | | | **II-1** | | | **III-1** | | | **CB** | | | **V-1** | | | **VI-1** | | | **VII-1** | | |
| BMS | −0.058 | **0.020** | 1.467 | −0.037 | **0.037** | 5.659 | **0.030** | **0.068** | 0.904 | **0.074** | **0.031** | 0.650 | **0.049** | **0.081** | 1.108 | **0.081** | **0.066** | **−0.197** | **0.058** | **0.111** | 2.429 |
| GBVS | −0.089 | **0.011** | 1.575 | −0.065 | **0.020** | 5.747 | −0.013 | **0.045** | 1.057 | **0.029** | **0.023** | 0.778 | −0.004 | **0.057** | 1.267 | **0.018** | **0.045** | **−0.021** | **0.008** | **0.081** | 2.601 |
| RARE2012 | −0.020 | **0.022** | 1.461 | −0.016 | **0.026** | 5.686 | **0.033** | **0.052** | 0.990 | **0.051** | **0.030** | 0.719 | **0.052** | **0.066** | 1.183 | **0.068** | **0.056** | **−0.109** | **0.062** | **0.083** | 2.515 |
| SIM | −0.024 | **0.023** | 1.435 | **0.002** | **0.041** | 5.530 | **0.092** | **0.069** | 0.796 | **0.116** | **0.032** | 0.591 | **0.095** | **0.080** | 1.008 | **0.119** | **0.065** | **−0.285** | **0.141** | **0.121** | 2.251 |
| SUN | −0.002 | **0.023** | 1.413 | **0.022** | **0.040** | 5.505 | **0.102** | **0.066** | 0.797 | **0.104** | **0.031** | 0.586 | **0.100** | **0.074** | 1.011 | **0.119** | **0.060** | **−0.268** | **0.144** | **0.112** | 2.257 |
| | **I-2** | | | **II-2** | | | **III-2** | | | **IV-1** | | | **V-2** | | | **VI-2** | | | **VII-2** | | |
| BMS | −0.005 | **0.028** | 0.103 | −0.145 | −0.011 | 3.403 | **0.025** | **0.066** | 0.933 | **0.060** | **0.069** | **−0.171** | **0.075** | **0.100** | 0.901 | **0.022** | **0.074** | 1.441 | **0.094** | **0.116** | 0.611 |
| GBVS | −0.039 | **0.022** | 0.198 | −0.168 | −0.024 | 3.477 | −0.016 | **0.044** | 1.080 | **0.009** | **0.046** | 0.004 | **0.019** | **0.071** | 1.082 | −0.035 | **0.046** | 1.639 | **0.038** | **0.083** | 0.811 |
| RARE2012 | **0.025** | **0.033** | 0.114 | −0.109 | −0.022 | 3.483 | **0.032** | **0.051** | 1.013 | **0.047** | **0.053** | **−0.060** | **0.068** | **0.078** | 1.004 | **0.030** | **0.055** | 1.522 | **0.083** | **0.090** | 0.732 |
| SIM | **0.002** | **0.026** | 0.087 | −0.080 | **0.003** | 3.686 | **0.083** | **0.066** | 0.832 | **0.110** | **0.070** | **−0.270** | **0.138** | **0.100** | 0.777 | **0.080** | **0.080** | 1.319 | **0.166** | **0.116** | 0.467 |
| SUN | **0.023** | **0.026** | 0.068 | −0.057 | **0.003** | 3.724 | **0.095** | **0.064** | 0.830 | **0.113** | **0.066** | **−0.259** | **0.139** | **0.092** | 0.785 | **0.088** | **0.073** | 1.332 | **0.167** | **0.108** | 0.477 |
| | **I-3** | | | **II-3** | | | **III-3** | | | **IV-3** | | | **V-3** | | | **VI-3** | | | **VII-3** | | |
| BMS | **0.039** | **0.087** | 1.613 | −0.054 | **0.045** | **−0.178** | −0.034 | **0.036** | 1.180 | −0.002 | **0.042** | 0.126 | −0.001 | **0.062** | 2.214 | **0.017** | **0.057** | 1.039 | **0.103** | **0.083** | **−0.287** |
| GBVS | −0.013 | **0.060** | 1.794 | −0.081 | **0.029** | **−0.088** | −0.077 | **0.018** | 1.320 | −0.033 | **0.029** | 0.240 | −0.055 | **0.038** | 2.372 | −0.038 | **0.034** | 1.219 | **0.042** | **0.057** | **−0.094** |
| RARE2012 | **0.052** | **0.069** | 1.680 | −0.028 | **0.029** | **−0.132** | −0.025 | **0.021** | 1.265 | **0.018** | **0.037** | 0.178 | **0.015** | **0.047** | 2.264 | **0.024** | **0.044** | 1.102 | **0.083** | **0.068** | **−0.179** |
| SIM | **0.099** | **0.092** | 1.490 | **0.014** | **0.054** | **−0.220** | **0.020** | **0.045** | 1.066 | **0.021** | **0.039** | 0.085 | **0.051** | **0.067** | 2.115 | **0.066** | **0.061** | 0.940 | **0.161** | **0.080** | **−0.395** |
| SUN | **0.108** | **0.085** | 1.492 | **0.034** | **0.051** | **−0.213** | **0.030** | **0.041** | 1.076 | **0.046** | **0.039** | 0.071 | **0.060** | **0.061** | 2.115 | **0.073** | **0.056** | 0.955 | **0.157** | **0.075** | **−0.379** |

Even if biases show gain for at least two metrics, this gain is rather insignificant. Thus, **results confirm those obtained on EyeTrackUAV2: biases convey meaningful saliency information**, however, combining them with HCs as defined here is not optimal.

## 4. Discussion

We would like to further discuss several points we dealt with all along with this paper.

First, we justified the made assumption of 1D Gaussian interpolation for marginal distributions. In the future, it could be interesting to learn different approximations of distributions and see the variations occuring on extracted biases. Several options can be implemented, such as learning a GMM or an alpha-stable Lévy distribution [93,94]. The latter distributions learn skewed distributions, which under particular conditions can express a Gaussian. Similarly, the extraction of local maxima on MDs parameters distributions could be replaced. Parameters of GMM or a mixture of alpha-stable Lévy distributions are good alternatives. However, the difference made by such sophisticated implementations is possibly insignificant, which explains our choice to keep this study technically sound and rather simple. Further improvements can be brought about in the future.

We also could have investigated a local extrema extraction to compute MD statistics. However, statistics would have been irrelevant in the case of low congruence between subjects. Assuming a 1D-Gaussian is less of a compromise than this solution.

We would like to stress that metrics for dynamic saliency rely on the average over the sequence. Consequently, scores may not be representative of the power of prediction of every frame. We also used the averaging process prior to t-SNE to compute features for sequences. This strengthens the distinction observed between types of imagery and within UAV content. Future studies are needed to evaluate if other strategies, such as the median or a combination of means and std of the distribution of scores, are more pertinent.

Finally, regarding the study on HCs filtering, other ways to improve HCs can be envisioned. For instance, a competition on the frame level of HCs and biases may provide better prediction scores. We decided not to include such a study here not to cover up the true message of our paper: proving we designed a relevant dictionary of biases.

## 5. Conclusions

Understanding people's visual behaviors towards multimedia content has been the object of extensive research for decades. The extension of visual attention considerations to UAV videos is though underexplored because of the infancy stage of the UAV imagery and the additional complexity to deal with temporal content.

Previous studies identified that UAV videos present peculiarities that modify gaze deployment during content visualization. We wonder if we can observe the differences between this imagery and typical videos saliency-wise. If so, it appears reasonable to further study UAV salience patterns and to extract biases that are representative of it. Ultimately, a dictionary of biases forms a reliable low-complexity saliency system. Its potential spans from real-time applications embedded on UAVs to providing the dictionary as priors to saliency prediction schemes.

The distinction between conventional and UAV content is developed through the extraction of valuable features, describing human saliency maps (HSMs) in a high-dimensional space. We designed hand-crafted features based on fixations dispersion, human saliency spatial and temporal structure, vertical and horizontal marginal distributions, and last based on GMMs. They were computed over each frame of the datasets DHF1K (400 videos) for typical videos and EyeTrackUAV2 (43 videos) for the unconventional imagery. We then run a machine learning algorithm for dimension-reduction visualization, namely the t-SNE, to observe differences between visual attention in the two types of content.

Our results justify why learning saliencies over conventional imaging to predict UAV imaging is neither accurate nor relevant. The separation between both imaging based on extracted characteristics pleads for dedicated processing. In addition to finding distinctions between types of content, the t-SNE revealed within UAV content differences. We thus apply the t-SNE on EyeTrackUAV2 alone to identify classes that represent sets of similar saliency patterns in sequences. We based this study on marginal distributions, which will prove useful to generate parametric content-specific biases from these clusters. A hierarchical clustering algorithm, specifically a dendrogram with a Ward criterion, sorts out sequences and, after thresholding, exhibits seven clusters of mainly balanced classes.

For each class, we proceed by extracting parameters that are needed to generate biases. From horizontal and vertical marginal distributions, we compute means and standard deviations, assuming they follow a 1D-Gaussian distribution. We gather all resulting statistics over time and cluster and compute their distributions. Such distributions express the likelihood of the center coordinates and std of a 2D-Gaussian in saliency patterns. Accordingly, we want to extract only prevailing values and implement the extraction of local maxima to define biases parameters.

We carry out with the creation of biases. We first reconstruct two 1D-Gaussian that will stand for horizontal and vertical distributions. By multiplying both, we will obtained the bias pattern. Biases are generated for all the combinations of previous parameters, leading to a number of 296 patterns. This number is reduced by keeping only most meaningful biases on clusters in terms of CC, SIM and KL and that outdo the center bias. The ensuing dictionary contains 21 patterns dowlable on our dedicated web-page (https://www-percept.irisa.fr/uav-biases/).

A qualitative analysis hints that videos shot with high altitude may present saliency patterns with high longitudinal variability. Besides, it seems that UAV saliency is centered width-wise. Moreover, some biases show center-bias alike patterns that suit more to UAV, as they have a reduced std in at least one dimension. We think this copes with the small size of objects in UAV videos. Quantitatively, biases surpass others on their respective cluster and achieve fair scores. All of the above is in line with our expectations for single image saliency patterns.

Finally, we conduct a benchmark over EyeTrackUAV2 to compare biases efficiency against baselines and handcrafted features saliency models (HCs). The study is extended to EyeTrackUAV1 to check the external validity of the dictionary and our conclusions. Overall, the dictionary presents some patterns with a high power of generalization, that overall and per cluster or sequence surpass CB, sometimes human means (HMs) and even more interesting HCs. Content-specific patterns were proved accurate and useful in appropriate contexts.

We further explored the potential of biases by using the dictionary as a set of filters. The outcome is that biases bring knowledge that is different from HCs. Note that the achieved gain is relatively low, particularly for the most-performing HCs. We thus recommend the use of biases on their own and not as filters. A future application that seems highly relevant is to feed dynamic saliency schemes with this dictionary of priors.

Overall, this study justifies the need for specializing salience to the type of content one deals with, especially UAV videos. The main outcome of the paper is the creation of a dictionary of biases, forming a low-complexity saliency system that is sound, relevant, and effective. It lays the ground for further advancements toward dynamic saliency prediction in specific imaging.

**Author Contributions:** Conceptualization, investigation, formal analysis, and writing—original draft, A.-F.P.; Supervision, L.Z. and O.L.M.; Writing—review and editing, A.-F.P., L.Z., and O.L.M. All authors have read and agreed to the published version of the manuscript.

**Funding:** The presented work is funded by the ongoing research project ANR ASTRID DISSOCIE (Automated Detection of SaliencieS from Operators' Point of View and Intelligent Compression of DronE videos) referenced as ANR-17-ASTR-0009.

**Conflicts of Interest:** The authors declare no conflict of interest.

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
