# Peer review of "Inferring Visual Biases in UAV Videos from Eye Movements"

_drones, doi:10.3390/drones4030031_

Round 1

Reviewer 1 Report

The paper deals with certain parts of the imagery analysis.  The paper includes many numerical proves of authors's proclamations as well as mention tests and benchmarks that served as a prove. Thanks to this, the paper closely describes the creation of a dictionary of biases which helps to shape proposed saliency sys. The outcomes of the paper can serve as a basics for further research in the field of imagery analyses. I recommend this paepr to be published.

Author Response

We would like to thank all reviewers for their insightful comments on the paper, as these comments led us to an improvement of the work. Our revisions reflect all reviewers’ suggestions and comments.

Overall, we changed extensively the structure of the paper to improve readability and to avoid repetitions. We clarified certain parts based on reviewers’ comments, and have added literature to better motivate our work and to emphasize its high influence on the UAV imagery field. We also have revised the entire text for English inaccuracies and typos.

You will find attached the new version of the paper with colored text that guides reviewers through the modifications. We added the sentences in red, while those in blue were shifted. Small corrections are not colored.

Reviewer1

The paper deals with certain parts of the imagery analysis.  The paper includes many numerical proves of authors's proclamations as well as mention tests and benchmarks that served as a prove. Thanks to this, the paper closely describes the creation of a dictionary of biases which helps to shape proposed saliency sys. The outcomes of the paper can serve as a basics for further research in the field of imagery analyses. I recommend this paepr to be published.

Thank you very much for your review. We are glad you appreciate our work.

Reviewer 2 Report

The authors are examining whether handcrafted saliency features are capable of differentiating typical and UAV content, using t-SNE, and they proceed with the extraction of a dictionary of visual attention biases for UAV images that can be used as a low-complexity saliency prediction system

The language and the writing style of the authors is good and they provide a detailed description of their intuition and the experiment development.

However, there are some important issues identified

  • There is no related work towards the aim of the paper. The references are generic regarding saliency, datasets and methodologies used in the paper. While the objective is pursued in a novel way, it is not clear if this can be achieved in different ways more efficiently.
  • The description of the work is too long, often describing in length things that are fairly standard (handcrafted features). Please try to make the text more concise.
  • From the text: "Based on this representation, we have noticed a significant difference in the deployment of visual attention between UAV and conventional sequences" The conclusions extracted from the t-SNE are not definitive given that with different parameters the situation could change dramatically the produced graph. What happens with different parameters? In any case, UAV clusters seem to spread across the whole graph and the conclusion is not straight forward. 

Author Response

We would like to thank all reviewers for their insightful comments on the paper, as these comments led us to an improvement of the work. Our revisions reflect all reviewers’ suggestions and comments.
Overall, we changed extensively the structure of the paper to improve readability and to avoid repetitions. We clarified certain parts based on reviewers’ comments, and have added literature to better motivate our work and to emphasize its high influence on the UAV imagery field. We also have
revised the entire text for English inaccuracies and typos.
You will find attached the new version of the paper with colored text that guides reviewers through the modifications. We added the sentences in red, while those in blue were shifted. Small corrections are not colored.

Reviewer2
The authors are examining whether handcrafted saliency features are capable of differentiating typical and UAV content, using t-SNE, and they proceed with the extraction of a dictionary of visual attention biases for UAV images that can be used as a low-complexity saliency prediction system The language and the writing style of the authors is good and they provide a detailed description of
their intuition and the experiment development.
However, there are some important issues identified There is no related work towards the aim of the paper. The references are generic regarding saliency, datasets and methodologies used in the paper.
Thank you for advising including further literature. We may have not imphasized enough how novel this process is and how biases are usually studied. We have included paragraphs in the introduction to tackle this lack of reference. We also have emphasized a research project, encountered after submitting the paper, which follows a similar pipeline as ours to classify shot types. We have added
several precisions to justify why we selected t-SNE and hierarchical clustering.
In introduction, we have added “Our aim is to produce a dictionary of biases possibly fed as priors to static or dynamic saliency prediction models. It is a widespread solution to improve saliency models, and it can take several forms. For instance, in [23], the dynamic two-streamed network includes a
hand-made dictionary of 2D centered-Gaussians which provides different versions of the center bias.
To the best of our knowledge, it is the first time that one empirically extracts a dictionary of biases for UAV videos.
Our biases extraction follows a similar pipeline than a current attempt to analyze saliency patterns in movie shots to explore cinematography dynamic storytelling [24]. Their pipeline includes the definition of features (saliency maps predicted by Deep Gaze II [25]), a dimension reduction operation through a principal component analysis, and finally clustering using K-means.
Regarding the UAV ecosystem, deep learning is a mainstream tool for numerous applications, i.e., automatic navigation of UAV [19, 26, 27], object [27] or vehicle tracking [28-30}, (moving) object detection [31, 32] under real-time constraints [33]. Some works combine both object detection and tracking [34-37] or implement the automation of aerial reconnaissance tasks [38]. However, only a minority of works take benefits from Regions of Interest (ROI) [27, 39], sometimes in real time [40], which is a first step towards considering visual attention. We believe that saliency, and in particular a dictionary of biases, will enable the enhancement of current solutions, even real-time applications.”
In Section 2.3 “The t-SNE is advantageous for two reasons: it enables visualization, and most importantly, it comprehensively considers all dimensions in its representation.”
In Section 2.3.2 “Hierarchical clustering is highly beneficial in our context as it is less constrained than K-means. There is no need for prior knowledge, such as a number of classes, number of elements in groups, or even whether batches are balanced.”

While the objective is pursued in a novel way, it is not clear if this can be achieved in different ways more efficiently.

To the best of our knowledge, it is the first time that one extracts a dictionary of biases empirically.
We are aware of a hand-made dictionary or the inclusion of attentive models in deep learning architecture. In addition, we are aware of unpublished work from the BBC that follows a similar idea and employed saliency maps as features, PCA for dimension reduction, and finally K-mean for clustering.
In our opinion, t-SNE is more accurate than PCA. t-SNE considers distances in the entire space, considering all components. PCA enables approximations to most important components. We thus argue t-SNE is more precise than PCA in our context.
Regarding the identification of classes, hierarchical clustering relies on less prior information than Kmeans, such as the number of classes, if classes are balanced, or the number of elements constituting a class. We have added these details to the paper, as you could read it above.

The description of the work is too long, often describing in length things that are fairly standard (handcrafted features). Please try to make the text more concise.
Thank you for pointing this out. The text is long indeed. We have removed pieces of information that are not important while kept the data necessary to make the study reproducible. We also tried our best not to repeat ourselves when it is avoidable.

From the text: "Based on this representation, we have noticed a significant difference in the deployment of visual attention between UAV and conventional sequences" The conclusions extracted from the t-SNE are not definitive given that with different parameters the situation could change dramatically the produced graph. What happens with different parameters?
We agree that the text was not clear enough. We have tried several parameters (perplexity: [2,5,10,15,20,30,50,100,150,200,500], various number of components and iterations and learning rate) and ensured to obtain similar results. Besides, it is not possible to obtain the same results with the same parameters; so even less with different ones. We thus run t-SNE several times per parameter settings as well.
It is not possible to report all the results. We thus decided to include illustrations with the same perplexity value for more readability. We have slightly changed the text accordingly. Unillustrated results also showed a distinction between imaging types and within UAV elements. See in section 2.3.1 “e.g. 2, 5, 10, 15, 20, 30, and 50” and “All results follow similar tendencies over all tested parameters.”
Regarding perplexity, values over 50 are not considered meaningful due to asymptotic behaviors.
Still, we have checked them to be sure. However, we do not report having tested these high values.

In any case, UAV clusters seem to spread across the whole graph and the conclusion is not straightforward.
Again, thank you for the useful comment. We updated the paper so that it encompasses further description or justification regarding the conclusion outlined from our results. It is actually because of the spread of points across the whole graph that we have decided later on to perform clustering.
We should read on graphs in Figure 2 that

(1) points representing UAV contents are in an area of the space not covered by typical points,
(2) there are various clusters of UAV points.
We detailed this outcome in the legend of figure 2 “We can see that most points representing UAV content are in areas of the space not covered by typical points and that there are various clusters of UAV points”
Should one point satisfy (1), it means that no typical content is embedding the same characteristics as this UAV video or frame. If most points of UAV content show (1), then we can deduce that we need to deploy UAV-specific analyses as typical contents can not express and are not representative of UAV saliency characteristics.
(2) means that we have sets of UAV videos or frames that exhibit similar patterns in the space defined by hand-crafted features. Each set embeds different and representative features of saliency maps that can be further explored to derive biases. This justifies we perform clustering.

Thank you very much for all your comments that contribute to enhancing the paper quality and facilitates the understanding of our work. We hope we have corrected the text faithfully to your review.

Reviewer 3 Report

The text provides am interesting approach to analyze human visual behavior on airborne sampled footage

Details

Table1: How are the features weighted?

  1. 182: „This represents an overall number of 275031 frames, 232790 for typical and 42241 for UAV content?” Longer sequences contribute more frames of the same content. How is that taken into account?

Figure 2: Please explain the axis labeling and scaling. Why is it different for 2a)?

  1. 248 „Human Means (HMs) thumbnails allow verifying the similarity between sequences HSMs in clusters.“ How? Please give an example.

Figure 4: alpha_r is doubled in the middle image

Genaral

Although the authors conclude that gaze behavior is dependent on image content,

(l. 42: “biases in UAV videos […] have the advantage to be content-specific”)

the study does not address the context in which the tasks were created. Sequences that were created to support the detection of objects differ in their nature (dynamics; contrast; sensor type; color schemes) fundamentally from images taken for large-scale surveillance or e.g. for tracking vehicles. The composition of the data sets in this respect would be interesting.

Footage taken from aircraft often differ in perspective and orientation due to the gimbal-mounted camera system. Screen-up is not always elevation-up. Therefore statements like

l. 389: "Besides, the 2D Gaussian is located slightly below the center of the image height-wise. No relation between this fact and sky presence was found based on the annotations of the EyeTrackUAV2 dataset."

should be made with caution.

There is also no report on the background of the human observers' tasks. What was the task they had while their eye movement was recorded? It can be assumed that the task had an influence on the gaze behaviour. The authors seem to suspect this:

(l. 272 „As we can see in Figure 1, this claim is reasonable if there is a single-object tracking (i.e., 1a and 1b) However, it is more questionable when there is no object of interest“ and

l. 397 “As already mentioned, the position of objects of interest, in this context in the upper part of the content, strongly influence visual attention” and

l. 589 “Overall, this study justifies the need for specializing salience to the type of content one deals with, especially UAV videos”)

, but did seemingly not consider it in their investigations.

Moreover, persons who work with drone imagery are not just amateurs. Often professional aerial photographers or sensor operators undergo special training, which in turn influences their visual routines.

Finally, often the repositioning of the sensor system is one of the tasks of such sensor or drone operators. This aspect, too, has the potential to shape their gaze behavious differently from that of an uninvolved observer.

It would be desirable if the authors would address these aspects.

Author Response

We would like to thank all reviewers for their insightful comments on the paper, as these comments led us to an improvement of the work. Our revisions reflect all reviewers’ suggestions and comments.

Overall, we changed extensively the structure of the paper to improve readability and to avoid repetitions. We clarified certain parts based on reviewers’ comments, and have added literature to better motivate our work and to emphasize its high influence on the UAV imagery field. We also have revised the entire text for English inaccuracies and typos.

You will find attached the new version of the paper with colored text that guides reviewers through the modifications. We added the sentences in red, while those in blue were shifted. Small corrections are not colored.

Reviewer 3

The text provides an interesting approach to analyze human visual behavior on airborne sampled footage

Details

Table1: How are the features weighted?

182: „This represents an overall number of 275031 frames, 232790 for typical and 42241 for UAV content?” Longer sequences contribute more frames of the same content. How is that taken into account?

Thank you for identifying this area of potential ambiguity. We do not perform features weighting. When dealing with sequences, we compute the standard deviation and average of features over the entire set of frames (fig 2 a). This gives us 76 features. That is, sequence results are independent of the number of frames. We state it now in section 2.2.5 “ This gives 76 features per video, independently of the sequences’ number of frames“.  Regarding frames, all features contribute to the analysis (fig 2 b,c). 

Figure 2: Please explain the axis labeling and scaling. Why is it different for 2a)?

We agree that scales are misleading. Axes in Figure 2 are that of the 2D space defined by the t-SNE. The values are not indicative as we don’t know the transform function computed by the t-SNE. In other words, there is no way to relate these numbers to feature values simply. We have decided to remove them to avoid ambiguity.

248 „Human Means (HMs) thumbnails allow verifying the similarity between sequences HSMs in clusters.“ How? Please give an example.

This verification is qualitative. We can verify in one glance the similarity between HMs in the same cluster. We have added a comment on cluster II that shows a common pattern: “For instance, in group II, there is a recurrent vertical and thin salient pattern.” Also, the example of group IV illustrates a heterogeneous group of HMs. We hope this is clearer.

Figure 4: alpha_r is doubled in the middle image

Thank you for the comment. It is an annoying typo indeed. We have made the necessary corrections.

 General

Although the authors conclude that gaze behavior is dependent on image content,

(l. 42: “biases in UAV videos […] have the advantage to be content-specific”)

the study does not address the context in which the tasks were created. Sequences that were created to support the detection of objects differ in their nature (dynamics; contrast; sensor type; color schemes) fundamentally from images taken for large-scale surveillance or e.g. for tracking vehicles. The composition of the data sets in this respect would be interesting.

The dataset, as well as the type of eye-tracking data collected, are of major importance. We have indicated that gaze information was collected under free-viewing conditions. That is, the experiment ground truth is based on no task.  We specified it in section 2.1 In this study, we consider only gaze information collected in free-viewing conditions. Observers can explore and freely appreciate the content.”

The dataset EyeTrackUAV2 provides both free-viewing and surveillance-based task.  It covers a wide range of different contents: it is compliant with object detection and contains videos with no salient object. However, we only used free-viewing information. We are then not dependent on task-specific biases. For further content properties, we leave the reader to refer to the dataset paper.  These details have been added to section 2.1.2 Besides, EyeTrackUAV2 was created in view to provide both free-viewing and surveillance-based task. There are indeed contents compliant with objects detection and tracking, and contents with no salient object.

Footage taken from aircraft often differ in perspective and orientation due to the gimbal-mounted camera system. Screen-up is not always elevation-up. Therefore statements like

  1. 389: "Besides, the 2D Gaussian is located slightly below the center of the image height-wise. No relation between this fact and sky presence was found based on the annotations of the EyeTrackUAV2 dataset."

We agree that screen-up and elevation-up are not necessarily the same. Our aim was not to suggest this statement. We wanted to relate the saliency horizontal shift -from the center to slightly above it- to content features. We removed this part as this comment does not rely on tangible proof and is not clear enough.

There is also no report on the background of the human observers' tasks. What was the task they had while their eye movement was recorded?

We do agree that, should there be a task, it would drastically affect gaze deployment and saliency. As we considered free viewing, we cannot interpret further the implication of a specific task.

 It can be assumed that the task had an influence on the gaze behaviour. The authors seem to suspect this:

(l. 272 „As we can see in Figure 1, this claim is reasonable if there is a single-object tracking (i.e., 1a and 1b) However, it is more questionable when there is no object of interest“ and

  1. 397 “As already mentioned, the position of objects of interest, in this context in the upper part of the content, strongly influence visual attention” and
  2. 589 “Overall, this study justifies the need for specializing salience to the type of content one deals with, especially UAV videos”)

, but did seemingly not consider it in their investigations.

The advantage in considering free-viewing is that viewers decide whether they perform object recognition, tracking, or if they extensively explore the content or do both. Therefore, we consider all types of attention and do not investigate further because our data do not allow a valid analysis of task-based attention.

Moreover, persons who work with drone imagery are not just amateurs. Often professional aerial photographers or sensor operators undergo special training, which in turn influences their visual routines.

Finally, often the repositioning of the sensor system is one of the tasks of such sensor or drone operators. This aspect, too, has the potential to shape their gaze behavious differently from that of an uninvolved observer.

It would be desirable if the authors would address these aspects.

You definitely have a point. The photographer bias is still present but takes a different form. We have updated the text to comment on this aspect. We believe that capture constraints and routines contribute to have different saliency patterns, as content characteristics do too. 

We have specified the above in introduction:” This imaging differs from conventional contents on various aspects. For instance, the photographer bias is not the same due to the special training and visual routines required to control the aerial sensor [18,19]. Shot images and videos represent objects under a new and unfamiliar birds’ perspective, with new camera and object motions, among others [20]. We thus wonder whether such variations impact visual explorations of UAV contents.”

Thank you very much for the constructive comments. We believe your remarks contribute to improve this paper. We have tried our best answering your concerns and applying corrections.

Reviewer 4 Report

This paper presents a process of extracting visual attention biases from UAV imagery. These visual biases are then analyzed with two datasets. The results show that the extracted biases can be used as a low-complexity saliency predication system.  I give my comments as follows:

  • The contribution of this paper to the respective field is unclear. After defining the visual biases in the introduction, the authors should further discuss comprehensively the other contributions to this field (literature studies) including those used the typical imagery.
  • The abstract should also include significant results from this paper that conclude the findings.
  • The authors need to read and check again the flow of the paper. In the first paragraph of Section 2, the authors claimed that they have introduced the typical and UAV datasets. However, the datasets were only introduced after that.
  • Section 2.2 which describes the handcrafted features needs to be restructured. When reading Section 2.2.1 to 2.2.4, the readers always need to refer back to some information in Section 2.2. The content seems to be disconnected. I suggest the authors only briefly describe the four representations in Section 2.2 and then their details in the following subsections.
  • Some abbreviations used without defining them when they are first introduced, e.g., DBSCAN, HDBSCAN, etc.
  • The sentence in between Eqs. (8) and (9) describes some symbols and operators used in these equations. However, they are some typos or mistakes in either the equations or the descriptions. For instance, [.] are missing in the equations? Or some symbols are missing in [.]?
  • Starting from Section 3 until 6, the authors combined the methodology and results together in the content. First, they are quite lengthy. Second, readers cannot distinguish the results are from this work or others as the discussion also includes some literatures. I recommend the authors to discuss ONLY your methodology/ contributions in those sections and move the literatures to your introduction.
  • Table 5-9 needs to be enlarged as they are currently too small.
  • The conclusion is too lengthy. The authors should only focus on their contributions/ results in this paper. The description of the future work can be shortened to one paragraph only.
  • Please modify the title by avoiding the repetition of words.

Author Response

We would like to thank all reviewers for their insightful comments on the paper, as these comments led us to an improvement of the work. Our revisions reflect all reviewers’ suggestions and comments.

Overall, we changed extensively the structure of the paper to improve readability and to avoid repetitions. We clarified certain parts based on reviewers’ comments, and have added literature to better motivate our work and to emphasize its high influence on the UAV imagery field. We also have revised the entire text for English inaccuracies and typos.

You will find attached the new version of the paper with colored text that guides reviewers through the modifications. We added the sentences in red, while those in blue were shifted. Small corrections are not colored.

Reviewer 4

This paper presents a process of extracting visual attention biases from UAV imagery. These visual biases are then analyzed with two datasets. The results show that the extracted biases can be used as a low-complexity saliency predication system.  I give my comments as follows:

    The contribution of this paper to the respective field is unclear. After defining the visual biases in the introduction, the authors should further discuss comprehensively the other contributions to this field (literature studies) including those used the typical imagery.

We understand your concern and a lack of literature is a bad flaw.

We have added a small paragraph on the research on the center bias, the most extensively studied bias in conventional imaging in the introduction. “When watching natural scenes on a screen, observers tend to look at the center irrespective of the content [1,2]. This center bias is attributed to several effects such as the photographer bias [2], the viewing strategy, the central orbital position [3], the re-centering bias, the motor bias [4,5], the screen center bias [1], and the fact that the center is the optimized location to access most visual information at once [1].  It is usually represented as a centered isotropic Gaussian stretched to the video frame aspect ratio [6,7]. Other biases are present in gaze deployment, mainly due to differences in context, i.e. observational task [8], population sample [9], psychological state [10], or content types [11–13]. Biases are studied through various modalities, such as reaction times for a task [8,9], specifically designed architecture [9] or through hand-crafted features describing salience or fixations [11]. Note that, except for the center bias, we are not aware of visual attention biases that take the form of saliency patterns.”

We rephrased how a dictionary of biases identified empirically is different from current solutions to enhance saliency models. “Our aim is to produce a dictionary of biases possibly fed as priors to static or dynamic saliency prediction models. It is a widespread solution to improve saliency models, and it can take several forms. For instance, in [23], the dynamic two-streamed network includes a hand-made dictionary of 2D centered-Gaussians which provides different versions of the center bias. To the best of our knowledge, it is the first time that one empirically extracts a dictionary of biases for UAV videos.

 Our biases extraction follows a similar pipeline than a current attempt to analyze saliency patterns in movie shots to explore cinematography dynamic storytelling [24]. Their pipeline includes the definition of features (saliency maps predicted by Deep Gaze II [25]), a dimension reduction operation through a principal component analysis, and finally clustering using K-means.”

We have added a section, still in the introduction, on current research on the UAV imaging that can benefit from this work (object recognition and tracking, navigation, and task automation).  “Regarding the UAV ecosystem, deep learning is a mainstream tool for numerous applications, i.e., automatic navigation of UAV  [19,27,28], object [28] or vehicle tracking [29–31], (moving) object detection [32,33] under real-time constraints [34].  Some works combine both object detection and tracking [35–38] or implement the automation of aerial reconnaissance tasks [39]. However, only a minority of works take benefits from Regions of Interest (ROI) [28,40], sometimes in real time [41],which is a first step towards considering visual attention. We believe that saliency, and in particular a dictionary of biases, will enable the enhancement of current solutions, even real-time applications.”

The abstract should also include significant results from this paper that conclude the findings.

We agree that more details can be included in the abstract. However, we restricted the description of our results to keep the abstract short to “we stress the differences between typical and UAV videos, but also within UAV sequences. “ and “We then conduct a benchmark on two different datasets, whose results confirm that the 20 defined biases are relevant as a low-complexity saliency prediction system.” To our mind, adding more information will require significantly more space and the abstract will not fulfill the conciseness constraint anymore.

The authors need to read and check again the flow of the paper. In the first paragraph of Section 2, the authors claimed that they have introduced the typical and UAV datasets. However, the datasets were only introduced after that.

Thank you for pointing out the issue of structure. We actually modified extensively the flow of the paper based on the comments of all reviewers and editors. The paper now follows the structure “Introduction”, “Material and Methods”, “Results”, “Discussion”, and “Conclusion”.

Regarding section 2, we believe this is a phrasing mistake. We blame the fact that we are not native. The formulation is valid in French but it could be misleading in English. We thus modified it to be correct and proper.

Section 2.2 which describes the handcrafted features needs to be restructured. When reading Section 2.2.1 to 2.2.4, the readers always need to refer back to some information in Section 2.2. The content seems to be disconnected. I suggest the authors only briefly describe the four representations in Section 2.2 and then their details in the following subsections.

Thank you for this constructive comment. We have modified the structure of the text as you suggest.

    Some abbreviations used without defining them when they are first introduced, e.g., DBSCAN, HDBSCAN, etc.

The acronyms were indeed poorly defined. We have modified the text so that it better introduces the two notions. We have not repeated the DBSCAN for the second acronym though, not to make the text heavy.   

The sentence in between Eqs. (8) and (9) describes some symbols and operators used in these equations. However, they are some typos or mistakes in either the equations or the descriptions. For instance, [.] are missing in the equations? Or some symbols are missing in [.]?

They are notations for different operators. They are not full brackets. Having a closed bracket above or below indicates the operator used. For more clarity and to reduce the text, we removed the equation that brought more issues than clarity to the text.

    Starting from Section 3 until 6, the authors combined the methodology and results together in the content. First, they are quite lengthy. Second, readers cannot distinguish the results are from this work or others as the discussion also includes some literatures. I recommend the authors to discuss ONLY your methodology/ contributions in those sections and move the literatures to your introduction.

We think about this work as a step-by-step process, not easy to separate while avoiding repetitive information. However, this structure does not help to understand our work. Consequently, as stated above, we have extensively revised the structure of the paper. We hope that the new structure will please you and increase the paper’s readability.   

Table 5-9 needs to be enlarged as they are currently too small.

Thank you for the comment, tables were indeed rather small for comfortable reading. We have made the necessary modifications.

    The conclusion is too lengthy. The authors should only focus on their contributions/ results in this paper. The description of the future work can be shortened to one paragraph only.

We have included a discussion section that shortens the conclusion. It also gives room to comments on the choices we made regarding our design. One of its advantages is to cut the part that should be removed from the conclusion.   

Please modify the title by avoiding the repetition of words.

We agree that this formulation is inelegant. We have modified the title to avoid unfortunate phrasing.

Overall, thank you very much for these on to the point comments. The paper has gone through a massive revision. Hopefully, that will increases its readability and scientific value.

Round 2

Reviewer 2 Report

Thanks to the authors for their extensive reply and changes performed. It has helped significantly to disambiguate the methodology and prove the merit of the method. I believe that this work can be published in the current form.

Reviewer 4 Report

Thank you for the correction.